# Loss of Septation Initiation Network (SIN) kinases blocks tissue invasion and unlocks echinocandin cidal activity against *Aspergillus fumigatus*

**Ana Camila Oliveira Souza**[1☯], **Adela Martin-Vicente**[1☯], **Ashley V. Nywening**[2], **Wenbo Ge**[1], **David J. Lowes**[1], **Brian M. Peters**[1,3], **Jarrod R. Fortwendel**[1,3]*

**1** Department of Clinical Pharmacy and Translational Science, College of Pharmacy, University of Tennessee Health Science Center, Memphis, Tennessee, United States of America, **2** Integrated Program in Biomedical Sciences, College of Graduate Health Sciences, University of Tennessee Health Science Center, Memphis, Tennessee, United States of America, **3** Department of Microbiology, Immunology, and Biochemistry, College of Medicine, University of Tennessee Health Science Center, Memphis, Tennessee, United States of America

☯ These authors contributed equally to this work.
* jfortwen@uthsc.edu

**Data Availability Statement:** All relevant data are within the manuscript and it Supporting Information files.

## Abstract

Although considered effective treatment for many yeast fungi, the therapeutic efficacy of the echinocandin class of antifungals for invasive aspergillosis (IA) is limited. Recent studies suggest intense kinase- and phosphatase-mediated echinocandin adaptation in *A. fumigatus*. To identify *A. fumigatus* protein kinases required for survival under echinocandin stress, we employed CRISPR/Cas9-mediated gene targeting to generate a protein kinase disruption mutant library in a wild type genetic background. Cell wall and echinocandin stress screening of the 118 disruption mutants comprising the library identified only five protein kinase disruption mutants displaying greater than 4-fold decreased echinocandin minimum effective concentrations (MEC) compared to the parental strain. Two of these mutated genes, the previously uncharacterized *A. fumigatus sepL* and *sidB* genes, were predicted to encode protein kinases functioning as core components of the Septation Initiation Network (SIN), a tripartite kinase cascade that is necessary for septation in fungi. As the *A. fumigatus* SIN is completely uncharacterized, we sought to explore these network components as effectors of echinocandin stress survival. Our data show that mutation of any single SIN kinase gene caused complete loss of hyphal septation and increased susceptibility to cell wall stress, as well as widespread hyphal damage and loss of viability in response to echinocandin stress. Strikingly, mutation of each SIN kinase gene also resulted in a profound loss of virulence characterized by lack of tissue invasive growth. Through the deletion of multiple novel regulators of hyphal septation, we show that the non-invasive growth phenotype is not SIN-kinase dependent, but likely due to hyphal septation deficiency. Finally, we also find that echinocandin therapy is highly effective at eliminating residual tissue burden in mice infected with an aseptate strain of A. *fumigatus*. Together, our findings suggest that

**Funding:** This work was supported by NIH/NIAID (https://www.niaid.nih.gov/) awards R21AI142509 (JRF), R21AI139388 (JRF), R01AI158442 (JRF), R21AI127942 (BMP) and R01AI134796 (BMP). The funders had no role in study design, data collection and analysis, decision to publish, or preparation of the manuscript.

**Competing interests:** The authors have declared that no competing interests exist.

inhibitors of septation could enhance echinocandin-mediated killing while simultaneously limiting the invasive potential of *A. fumigatus* hyphae.

## Author summary

*Aspergillus fumigatus* is a ubiquitous fungal pathogen and the major causative agent of a life-threatening infection known as invasive aspergillosis (IA). IA is a disease in which, typically immune compromised, patients experience unfettered growth of this pathogenic mold in the lungs with dissemination to other organs being common. Therapeutic options are limited by multiple factors and only three major antifungal drug classes are currently in use. Although effective as treatment for other fungal diseases, the echinocandin class of antifungals have limited usefulness against IA. Our overall goal is to identify novel fungal proteins that, if targeted for inhibition as part of a co-therapeutic approach, could improve the anti-*Aspergillus* activity of echinocandins and therefore lead to better patient outcomes. Here, we sought to identify *A. fumigatus* genes required for fungal survival under echinocandin stress by generating and screening an *A. fumigatus* mutant library composed of disruptions in genes encoding putative protein kinases. We found that protein kinases required for hyphal septation are essential for survival in the presence of echinocandins and, surprisingly, for the ability of *A. fumigatus* to invade lung tissue. Our results suggest that novel septation inhibitors could enhance echinocandin activity while simultaneously limiting *A. fumigatus* virulence.

## Introduction

*Aspergillus fumigatus* is among the most common causes of human invasive fungal infections in immunocompromised individuals, including solid organ transplant recipients, those undergoing hematopoietic stem cell transplant, and patients receiving highly immunosuppressive chemotherapies [1–3]. If untreated, these infections are almost always fatal and, even with proper diagnosis and treatment, are associated with an overall ~50% mortality rate [4]. Furthermore, the estimated annual cost of *Aspergillus* infections in the U.S. approaches $1 billion [5]. The most life-threatening *Aspergillus* infection occurs typically in the setting of profound and prolonged immune suppression and is known as invasive aspergillosis (IA). IA is initiated by the inhalation of *A. fumigatus* conidia from the environment [6]. In the immune compromised host, these conidia undergo a process of germination characterized by an initial phase of isotropic swelling followed by a switch to highly polarized growth leading to the formation of a germ tube. These germ tubes continue to extend through focused growth at the cell apex to generate the invasive hyphal forms that can invade surrounding tissue in search of nutrients, eventually reaching the pulmonary microvasculature system to disseminate [6]. Although decades of research have focused on *A. fumigatus* conidial adherence to and nutrient utilization in the host lung environment, as well as on the cellular and molecular processes essential for subsequent hyphal formation and invasion, our understanding of these processes remain incomplete.

Therapy of invasive aspergillosis is limited to three currently available classes of antifungal compounds. The polyene class, of which Amphotericin B is the only member used for invasive disease, can be associated with acute infusion-related toxicities as well as nephrotoxicity with prolonged administration [7]. The triazole class are the frontline treatment for *Aspergillus*

infections, with voriconazole considered the treatment of choice for this indication [8]. Treatment of aspergilloses is often prolonged, and mold-active antifungal prophylaxis employing triazole drugs is now common [9–13], both of which increase the potential for the development of drug-resistant organisms. Since the 1990s, triazole resistance in clinical isolates of this fungal pathogen has been increasing worldwide and is now the subject of significant research in the US and abroad [14–20]. Therefore, clinical use of the polyene and triazole classes is limited by patient toxicity and threatened by resistance, respectively.

The third major class of antifungals with anti-*Aspergillus* activity are the echinocandins, including caspofungin, micafungin and anidulafungin. These compounds are generally well-tolerated and are often used in salvage therapy for invasive infections [21]. Echinocandins are specific inhibitors of cell wall biosynthesis in fungi, as they inhibit the activity of the β-1,3-glucan synthase enzyme. This enzyme is encoded by the *fksA* gene in *A. fumigatus* and is the sole protein driving synthesis of the major cell wall component, β-1,3-glucan [22]. Whereas the activity of the echinocandins is fungicidal for the major yeast pathogens of the *Candida* genus, they are considered fungistatic against the *Aspergilli* [21,23]. Treatment of *A. fumigatus* with echinocandins causes lysis of hyphal tips and blunting of hyphal growth, but viability is maintained [24]. In addition, a caspofungin paradoxical effect (CPE) of growth inhibition has been described for caspofungin both *in vitro* and *in vivo* and is characterized by decreasing effectiveness of drug with increasing concentrations [21,25]. Current research suggests that the CPE is the result of the induction of tolerance mechanisms within hyphal compartments that survive caspofungin therapy. These mechanisms include remodeling of the cell wall, upregulation of cell wall integrity machinery and the induction of calcium-regulated stress pathways [21,25]. Although not conclusive, multiple studies using models of invasive aspergillosis have suggested that the CPE is not merely an *in vitro* phenomenon and may be an issue underlying treatment failure during caspofungin therapy of invasive aspergillosis in specific cases [25]. Likely underpinned by the fungistatic nature of the echinocandins against *Aspergilli*, breakthrough infections during echinocandin prophylaxis have been reported to be as high as 28% [26] and one study has identified echinocandin prophylaxis as an independent risk factor for breakthrough infections when compared with triazole prophylaxis [27]. Therefore, the echinocandins are mostly utilized where triazole therapy is contraindicated or has failed for invasive aspergillosis.

Recent studies have shown that the phospho-proteome of *Aspergilli* is highly responsive to echinocandin-induced stress, implying extensive kinase- and phosphatase-mediated re-wiring of cellular physiology for survival during inhibition of β-1,3-glucan biosynthesis [28–30]. Further, multiple studies in *A. fumigatus* have implicated protein kinase and protein phosphatase activity as important to cell wall stress imposed by echinocandins [31–36]. Together, these reports suggest that the further study of phospho-regulatory events required for survival during echinocandin-induced stress could uncover novel avenues for combination therapies directed at enhancing echinocandin activity against *Aspergilli* and other human pathogenic fungi. Here, we utilized a CRISPR/Cas9-based rapid gene disruption technique to generate a protein kinase gene disruption library in a wild type genetic background of *A. fumigatus*. Screening of this library for cell wall stress and echinocandin sensitivity phenotypes uncovered multiple protein kinases contributing to growth under each condition. In addition to the previously characterized cell wall integrity pathway and cAMP-mediated signaling protein kinases, our screens identified orthologs of the Septation Initiation Network (SIN) kinases as essential for growth under echinocandin-induced stress [37,38].

As septa are considered essential for the limitation of cell wall damage to filamentous fungal hyphae and the putative SIN is unstudied in *A. fumigatus*, we sought to further characterize the importance of each *A. fumigatus* SIN kinase to survival under echinocandin-induced stress both *in vitro* and during invasive disease. Our data indicate that each SIN kinase is essential for

septum formation and for survival under echinocandin-induced cell wall stress. Strikingly, each of the SIN kinase disruption mutants were avirulent in a corticosteroid model of invasive aspergillosis (IA) and two of the three mutants were also avirulent in a chemotherapeutic model of IA. This lack of virulence was characterized by loss of tissue invasion and inability to accumulate fungal burden. Nevertheless, using culture-based residual fungal tissue burden as a gold-standard determination of fungicidal activity, we show that echinocandin therapy was enhanced in mice infected with SIN kinase mutants. Further, we show that loss of additional regulators of septation also results in avirulence characterized by lack of tissue invasion and loss of viability under echinocandin stress, suggesting that our phenotypes are likely due to loss of septation and not septation-independent functions of the *A. fumigatus* SIN.

## Results

### Generation of an *A. fumigatus* protein kinase disruption mutant library

To identify protein kinase-driven pathways important for survival under echinocandin stress in *A. fumigatus*, we first generated a protein kinase disruption library in the A1163 (CEA10) wild type genetic background through coupling of CRISPR/Cas9-based gene targeting with a miniaturized protoplast transformation technique. Putative protein kinase genes were first identified through BLAST searches of the *A. fumigatus* A1163 (CEA10) genome database at FungiDB (fungidb.org) using the previously published known protein kinases of *Aspergillus nidulans* [39]. This search yielded 148 putative protein kinases representing 10 different protein kinase classes, as well as putative kinases falling into no known kinase class (S1 File). Of these 148 putative kinase genes, 142 were found to be encoded in the genomes of both sequenced laboratory strains, A1163 (CEA10) and Af293, and were therefore selected for disruption (S1 File). For library construction, we employed a miniaturized version of a CRISPR/Cas9-based gene editing technique, previously adapted in our lab, that provides up to 90% gene targeting efficiency in *A. fumigatus* [40,41]. Protospacer adjacent motif (PAM) sites for Cas9-induced double strand breaks and integration of hygromycin repair templates were selected using the Eukaryotic Pathogen CRISPR Guide RNA/DNA Design Tool (EuPaGDT, grna.ctegd.uga.edu) through batch upload analysis of all protein kinase coding sequences. Each PAM site was selected to direct double-strand DNA breaks near the putative transcriptional start site of each gene and integration of repair templates designed to disrupt readthrough of the first exon, when possible (S1 File). Cas9-ribonucleotides (RNPs) for gene targeting were assembled *in vitro* using custom designed guide RNAs (gRNAs) and commercially available Cas9 enzyme, as previously described [40]. Transformations were miniaturized into single wells of 96-well plates with a final well volume of 200 µl (Fig 1A). After transformation, total contents of individual wells were plated to osmotically stabilized agar and overlaid with hygromycin-containing top agar for selection (Fig 1B). Individual transformants were isolated to secondary selection plates and subsequently screened by multiple PCR reactions to confirm correct integration of the hygromycin repair template (Fig 1C). After three rounds of transformations, successful disruption of 118 protein kinase genes were confirmed (S1 File). The remaining 24 protein kinases for which disruptions were not achieved are largely orthologs of putatively essential genes in *A. nidulans*, suggesting conserved essentiality in *A. fumigatus* [39].

### Protein kinase-mediated regulation of *A. fumigatus* growth and asexual development

Of the 118 disrupted protein kinase mutant strains generated in this study, twenty-six were unable to grow at the same rate as the parental strain when cultured on standard laboratory

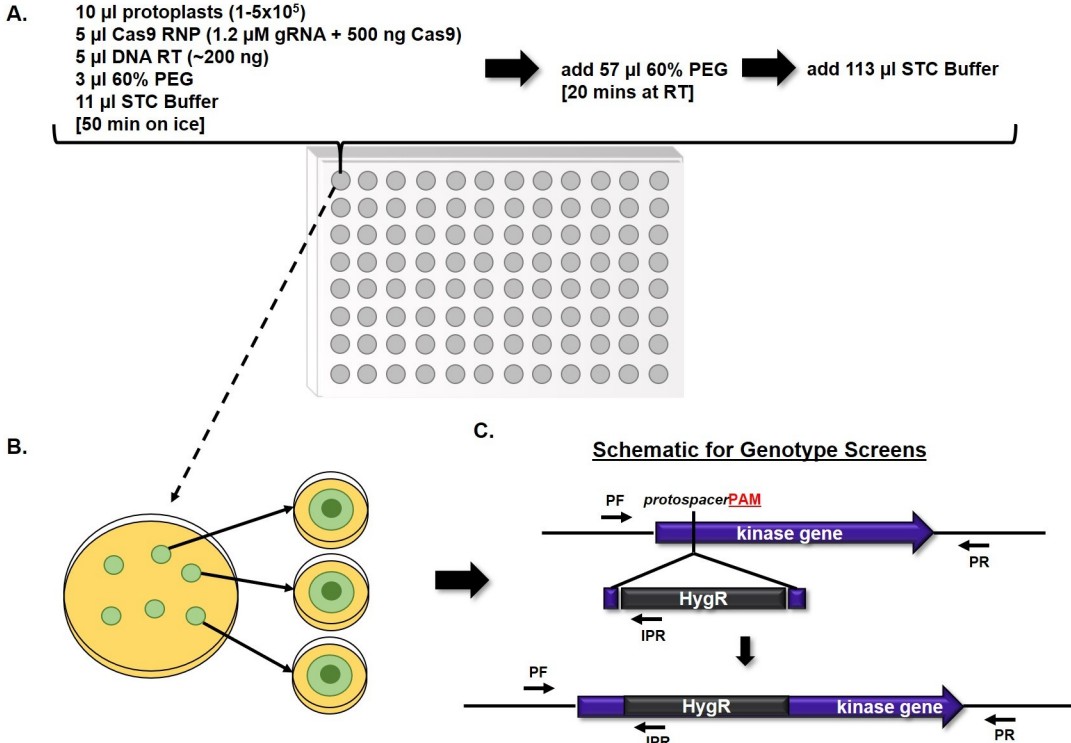

**Fig 1. Construction of a protein kinase disruption library in *A. fumigatus* by CRISPR/Cas9-mediated gene editing. A)** Miniaturized protoplast transformations were carried out in 96-well plates, with a final total volume of 200 µl per well, and each well representing an attempted disruption of a single protein kinase gene. **B)** After the transformation process, the entire contents of each well were spread onto individual sorbitol minimal medium (SMM) agar plates and allowed to recover overnight at room temperature before overlaying with hygromycin-containing top agar for selection. Following these transformation procedures, typically 10 to 30 transformant colonies were evident on each selection plate after 3 to 4 days of incubation at 37˚C. However, due to the high efficiency of gene targeting with the CRISPR/Cas9 system, only 3 to 4 colonies per transformation were required to be isolated for genotypic screening. **C)** Putative transformants were subjected to genotypic analyses by PCR to confirm proper integration of the repair template for gene disruption. These PCR analyses included screens with allele specific primer sets PF/PR and PF/IPR, pictured above. PF = Forward screening primer. PR = Reverse screening primer. IPR = Internal reverse screening primer complementary to HygR sequence. HygR = Hygromycin Resistance cassette, utilized as the repair template for gene disruption. All kinase genes were targeted for disruption at the 5' end of the gene (within the first exon, where possible), as indicated by the placement of the protospacer and protospacer adjacent motif (PAM, red bold font) above.

minimal media. Disruption of seven different protein kinases resulted in a reduction of growth of greater than 50% when compared to the parent strain, generating compact colonies that were unable to expand radially on minimal media (Figs 2A and S1A). Among these kinase disruptions were the cell wall integrity mitogen-activated protein kinase (MAPK), *mpkA* (AFUB_070630), and the upstream MAPK kinase (MAPKK), *mkkA* (AFUB_006190). Loss of either of these kinases has been previously shown to result in compact colony morphology [35]. Although deletion of the cell wall integrity MAPKK kinase (MAPKKK), *bck1* (AFUB_038060), was previously shown to result in reduced growth as well, disruption of this kinase in our library was associated with only a mild reduction in colony growth (Figs 2B and S1A). In addition, significantly reduced growth was generated by disruption of the catalytic subunit of Protein Kinase A, *pkaC1* (AFUB_027890), or the PAK-kinase, *cla4* (AFUB_053440) (Figs 2A and S1A). Both kinases were also previously characterized as important for vegetative growth in *A. fumigatus* [42,43]. Previously uncharacterized *A. fumigatus* protein kinases causing > 50% growth reduction upon disruption included orthologs of the eukaryotic

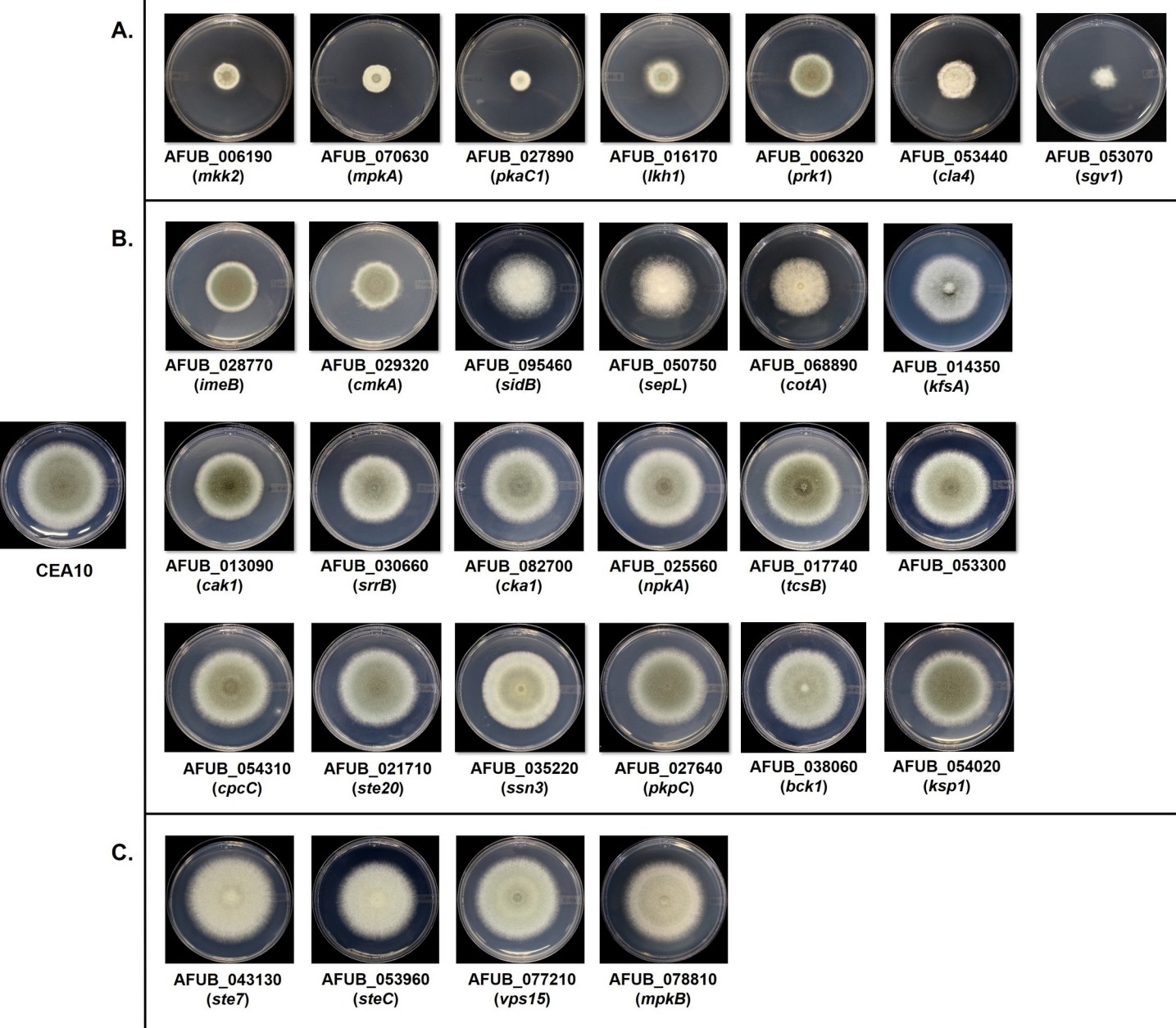

**Fig 2. Colony morphologies of selected *A. fumigatus* protein kinase disruption mutants.** 96-hr colony morphologies of severely (**A**) and moderately (**B**) growth restricted protein kinase disruption mutants, as well as colony morphologies of mutants that are not growth restricted but display reduced conidiation (**C**). Ten thousand conidia from each strain were point inoculated onto the center of minimal media agar and cultured for 96 hrs at 37˚C.

LAMMER kinase (*lkh1*; AFUB_016170), an *S. cerevisiae* kinase regulating the actin cytoskeleton (*prk1*; AFUB_006320), and a cyclin-dependent protein kinase (*sgv1*; AFUB_053070) (Figs 2A and S1A). Eighteen additional protein kinase disruptions resulted in mild-to-moderate growth reductions ranging between 10–50% of the parental strain (Figs 2B and S1A). Of the previously characterized kinase genes fitting into this category, we identified slow growth in disrupted orthologs of a phosphorelay sensor kinase (*tscB*; AFUB_017740) [44], the Cross-Pathway Control kinase (*cpcC*; AFUB_054310) [45], a p21-Activated Kinase (PAK) family

protein (*ste20/pakA*; AFUB_021710) [32], a cyclin-dependent protein kinase (*ssn3*; AFUB_035220) [46], and the cell wall integrity MAP kinase kinase kinase (*bck1*; AFUB_038060) [35] (Figs 2B and S1A). Thirteen mutant strains also displayed a significant impairment in asexual differentiation, as evidenced by significantly reduced conidia production, with nine of these kinase disruptions resulting in severe loss of conidiation when compared to CEA10 (Figs 2C and S1B). Six of these kinases have been previously characterized as required for conidiation, including three that comprise an asexual developmental kinase cascade in *A. fumigatus* (*steC*: AFUB_053960), *ste7/mkkB*; AFUB_043130, and *mpkB*; AFUB_078810) [47]. The remaining previously characterized kinases were *pkaC1*, *mpkA*, and *cla4*, each of which negatively impact asexual development upon deletion [35,42,43]. The finding that disruption of previously characterized growth-mediating kinases resulted in growth retardation of strains in our library supported the validity of our gene disruption approach. Therefore, these initial studies identified multiple novel protein kinases regulating *A. fumigatus* growth and development.

## Multiple protein kinases are required for growth under cell wall and echinocandin stress

To identify protein kinases required for cell wall stress tolerance, we performed spot-dilution assays in the presence of the common cell wall stress agents calcofluor white (CFW) and congo red (CR) on minimal media (MM), as well as caspofungin minimum effective concentration (MEC) assays, for all 118 viable kinase disruption mutants. Whereas caspofungin is an echinocandin-class antifungal that directly inhibits the fungal β-glucan synthase, CFW and CR are known to interfere with cell wall assembly by interacting with nascent chitin chains to prevent crosslinking of chitin to glucan moieties [48]. Spot-dilution assays identified seven protein kinase disruptions that displayed increased susceptibility to both CFW and CR, and an additional seven mutants that were hypersusceptible to only CR (Fig 3A and 3B). Among those 14 mutants found to be hypersusceptible to either cell wall active compound were the disruptions of the cell wall integrity kinases *mkkA* and *mpkA* (Fig 2A) which have previously been shown to be required for survival under various forms of cell wall stress [35]. Interestingly, the MAPKKK at the head of the *A. fumigatus* cell wall integrity pathway, Bck1, was again not identified by our assays as producing a cell wall stress hypersensitivity phenotype upon disruption. This finding, coupled with the lack of a severe growth restriction phenotype in Fig 2 for the *bck1* disruption mutant, indicated that some mutations in our library may be either non-disruptive or only partially disruptive to gene function. Our cell wall stress screens also uncovered protein kinases whose disruption generated resistance to either CFW (*pkaC1*) or to CR (*kfsA*, *cmkA*, and *stk22*), as evidenced by the increased ability to sustain colony formation under stress (Fig 2C).

To see if the CFW and CR susceptibility phenotypes correlated with echinocandin susceptibility, modified caspofungin MEC analyses were performed by broth microdilution (BMD) [49]. Whereas only five protein kinase disruption mutants were identified to display ≥ 4-fold increased susceptibility to caspofungin (i.e., at least two dilution shift), an additional 44 mutants displayed a 2-fold (one dilution) increase in caspofungin susceptibility (S1 File). Of those mutants that we previously identified as hypersusceptible to CFW, CR, or both, only the AFUB_087120, AFUB_013090 (*cak1*), and AFUB_018600 (*pom1*) disruption mutants showed no shift in caspofungin MEC values. All other cell wall stress susceptible mutants displayed at least a 2-fold reduction in caspofungin MEC values (S1 File). Interestingly, the disruption mutants for AFUB_09320 (*cmkA*), AFUB_014350 (*kfsA*), and AFUB_027890 (*pkaC1*) which displayed increased resistance to either CFW or CR, also displayed increased susceptibility to

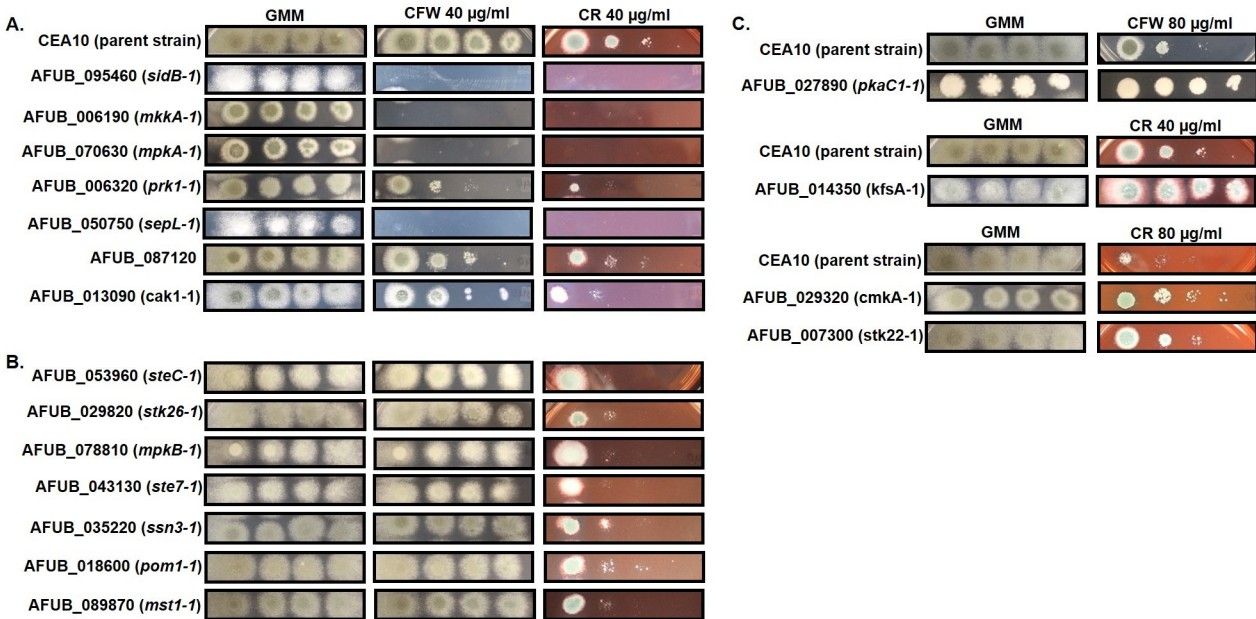

**Fig 3. Multiple protein kinases contribute to cell wall integrity in *A. fumigatus*. A**) Protein kinase gene disruption mutants displaying increased susceptibility to both cell wall disrupting agents, calcofluor white (CFW) and congo red (CR), by spot-dilution assay when compared to the wild type parent (CEA10). **B**) Protein kinase gene disruption mutants displaying hyper-susceptibility to only CR when compared to the parent strain. **C**) Protein kinase gene disruptants displaying increased resistance to CFW (*pkaC1-1*) or to varying concentrations of CR (*kfsA-1*, *cmkA-1*, and *stk22-1*). For each target protein kinase gene, the systematic name is listed with the strain name given in parentheses. Strain names were designed using either the previously published or putative (based on homology to *Aspergillus nidulans*) gene names with the addition of "-1" to indicate a disruption mutation of that gene. GMM = glucose minimal media with no CFW or CR added. For all assays, conidial inocula were applied at $10^4$, $10^3$, $10^2$, and $10^1$ total conidia and plates were incubated at 37°C for 72 hrs.

caspofungin (2-fold reduced MEC). Importantly, among the caspofungin hypersusceptible kinase mutants that displayed ≥ 4-fold decreased MEC values were those known to be involved in cell wall integrity signaling (*mpkA* and *mkk2*) and the cAMP-activated protein kinase (*pkaC1*), of which *mpkA* and *pkaC1* have been previously characterized as necessary for response to echinocandin stress [50,51].

Mutation of two additional kinases that are not members of the cell wall integrity pathway, AFUB_095460 (*sidB*) and AFUB_05070 (*sepL*), also displayed high levels of susceptibility in both of our cell wall stress and caspofungin MEC assays (Fig 3A and S1 File). These putative *A. fumigatus* kinases are orthologous to the *A. nidulans* SepL and SidB kinases that function as members of the Septation Initiation Network (SIN) kinase cascade. The core of the *A. nidulans* SIN pathway is composed of three kinases: the SteK-class proteins, SepH and SepL, and the AGC-class kinase, SidB (Fig 4A) [39]. Deletion of any single *A. nidulans* SIN kinase gene results in aseptate hyphae and reduction of conidiation whereas analysis of the *Neurospora crassa* SIN kinase orthologs has found only two of the three conserved kinases to be essential for the process of hyphal septation [39,52]. As the SIN pathway is completely uncharacterized in *A. fumigatus*, we sought to examine the importance of each SIN kinase to hyphal septation and to protection against echinocandin damage both *in vitro* and *in vivo* during invasive aspergillosis.

## *A. fumigatus* SIN kinases are required for septation and for survival in response to echinocandins

Although a disruption of the *sepH* gene was generated as part of our initial library construction, the *sepH-1* mutant did not show the same colony growth, conidiation, cell wall stress or

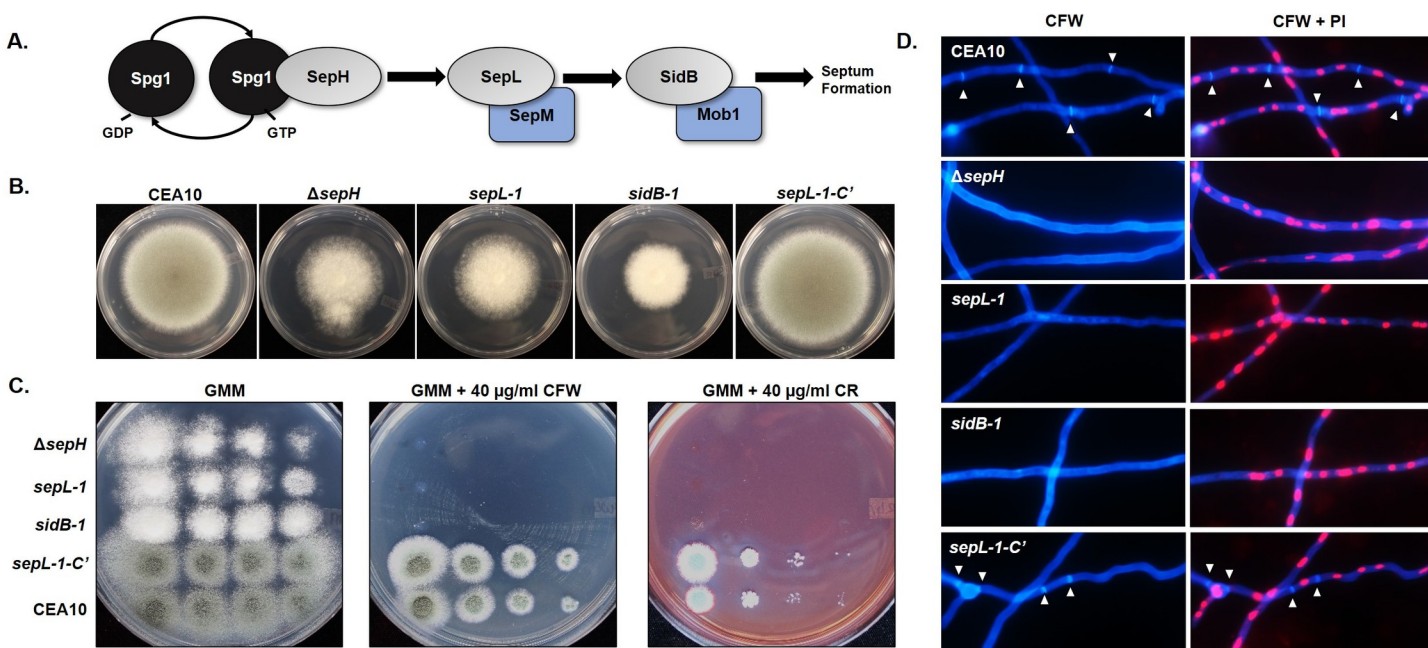

**Fig 4. The Septation Initiation Network (SIN) kinases are each required for hyphal septation and protection against cell wall damage in *A. fumigatus*. A)** The putative core SIN pathway in *A. fumigatus* based on signal transduction models constructed for *Schizosaccharomyces pombe* and *Aspergillus nidulans*. A protein kinase cascade, initiated by activation of the SepH kinase through interaction with the GTP-bound GTPase, Spg1, leads to downstream activation of the SepL and SidB kinases to eventually promote initiation of septation. SepL and SidB are shown with their putative regulatory binding partners, SepM and Mob1, respectively. **B)** Deletion of *sepH* (Δ*sepH*) phenocopies *sepL* and *sidB* disruption (*sepL-1* and *sidB-1*, respectively) as evidenced by restricted colony size and loss of conidiation (i.e., white colony formation). Complementation of SIN activity in the *sepL-1* disruption mutant by gene replacement (*sepL-1-C'*) results in full growth recovery and conidiation. Ten thousand conidia from each strain were spot-inoculated onto the center of a GMM agar plate and cultured for 96 hrs at 37°C. **C)** Loss of any single SIN kinase results in absence of growth in the presence of the cell wall destabilizing compounds CFW or CR. Conidia from each strain were spot inoculated in descending concentrations onto GMM alone or GMM containing either 40 µg/ml CFW or CR. **D)** Loss of any single SIN kinase results in the absence of septa in mature hyphae. Conidia from each strain were cultured to mature hyphae (16 hrs at 37°C) and subsequently stained with calcofluor white (CFW) and propidium iodide (PI) to visualize septa and nuclei, respectively. White arrowheads indicate septa in the CEA10 (parent) and *sepL-1* complemented (*sepL-1-C'*) strains. No septa were evident in the Δ*sepH*, *sepL-1* or *sidB-1* mutants.

echinocandin stress phenotypes as the *sepL-1* or *sidB-1* mutants. To see if our *sepH-1* mutant actually represented a loss of gene function, we also generated a complete gene deletion of *sepH* by CRISPR/Cas9 gene targeting (S2A and S2B Fig) and found that this mutant (Δ*sepH*) phenocopied the other SIN kinase disruption mutants with respect to colony phenotype and cell wall stress imposed by CFW and CR (Fig 4B and 4C). This finding indicated that, as for the *bck1-1* mutant, disruption of *sepH* using our approach likely only partially impacted function. For that reason, the Δ*sepH* deletion mutant and not the *sepH-1* disruption mutant was used moving forward. Inspection of mature hyphae from each mutant using CFW fluorescence staining to highlight the cell wall revealed that septation was completely ablated upon loss of any single SIN kinase (Fig 4D). Nuclear morphology and positioning in the each of SIN kinase mutants was grossly normal (Fig 4D), indicating that septation is not required for this aspect of *A. fumigatus* hyphal biology. This finding is in line with data reported for other septate filamentous fungi in which the cell cycle and cytokinesis (i.e., septation) are linked but not essentially coupled, as in yeast organisms [53]. Repair of SIN kinase pathway activity through complementation of the *sepL-1* disruption by re-integration of the *sepL* wild type allele into the native locus (S2C and S2D Fig) resulted in complete recovery of all growth and cell wall stress phenotypes (Fig 4B and 4C). In addition, this *sepL-1* complement strain (*sepL-1-C'*) displayed a complete recovery of septum formation (Fig 4D).

We next performed both E-test and fluorescence-based quantitative killing assays comparing the CEA10 parent and each SIN kinase mutant. E-test assays were performed utilizing both caspofungin (CAS) and micafungin (MFG) embedded strips and the production of a zone-of-clearance recorded after 48 hours of culture. In keeping with the fact that echinocandins are fungistatic against *Aspergilli*, the CEA10 parental strain displayed no zone-of-clearance surrounding E-test strips from either echinocandin (Fig 5A and 5B). Only an elliptical zone of depressed growth was evident for CEA10 and agar cores taken from within this zone grew normally when supplanted to agar containing no echinocandin (Fig 5A and 5B, inset). In contrast, each of the SIN kinase mutants developed a visible zone-of-clearance for both echinocandins suggesting significantly inhibited growth (Fig 5A and 5B). The only exception to this was found on the Δ*sepH* E-test plates for both echinocandins where a small number of compact microcolonies were able to form in the zone-of-clearance (Fig 5A and 5B, Δ*sepH* panels). Agar cores removed from the zone-of-clearance from each SIN kinase mutant plate were inviable when supplanted to drug-free agar. When specifically selected for sub-culture to drug-free agar, the Δ*sepH* zone-of-clearance microcolonies generated viable colonies but maintained hypersensitivity to both echinocandins on retest. These findings implied that the zone-of-clearance generated on the SIN kinase mutant E-test plates represented fungicidal activity of both echinocandins. In contrast, E-test and broth microdilution assays revealed no differences in voriconazole antifungal drug susceptibility among the SIN kinase mutants when compared to the parental strain (MIC = 0.5 μg/ml for all strains).

To more quantitatively measure SIN kinase mutant death in response to echinocandin stress, we also employed fluorescence-based assays using the live-cell stain 5-carboxyfluorescein diacetate (CFDA). CFDA is a cell-permeable esterase substrate that has been previously used as a viability indicator for *A*. *fumigatus* and other fungi [23]. Conidia from the parent strain and each SIN kinase mutant were grown in the presence or absence of 0.5 μg/ml micafungin and subsequently stained with CFDA to detect germlings and / or microcolonies with live hyphae or hyphal segments. Using fluorescence microscopy, individual microcolonies from each strain were scored as either live (CFDA-positive) or dead (CFDA-negative). In the absence of micafungin, the CEA10 parent and SIN kinase mutants displayed similar levels of CFDA-positive staining after 12 hours of culture, indicating similar baseline viability among the strain set (Fig 5C). Quantitation of CFDA-positivity past the 12-hour timepoint was not possible in the untreated samples, as continued hyphal growth obscured individual microcolonies for all strains. After 12 hours of growth in the presence of micafungin, each of the SIN kinase mutants displayed significant reductions in CFDA positivity, showing between only 40% and 60% positivity (Fig 5D). The CEA10 parent maintained an almost 90% positivity in staining. At the 12-hour timepoint, germlings in the CEA10 parent were noted to be either fully CFDA-positive (Fig 5E, white arrowhead) or to display unstained hyphal tip regions with positively stained subapical segments (Fig 5E, black dotted line arrows). In contrast, each of the SIN kinase mutants displayed either wholly CFDA-stained or -unstained (Fig 5E, black arrowhead) germlings. These findings suggested protection of interseptal hyphal segments of the CEA10 parent that was lost in the SIN kinase mutants due to lack of septation. Further, after 24 hours of culture in the presence of micafungin, the SIN kinase mutants displayed no CFDA positivity, whereas the CEA10 parent maintained almost 100% positive staining (Fig 5D).

Both the E-test and CFDA-based killing assays described above measured the impacts of echinocandin stress when applied to each strain from the onset of growth (i.e., before conidial germination begins). As septa were completely lacking in mature hyphae of each of the SIN kinase mutants, we also reasoned that damage from mature hyphal tip lysis induced by echinocandins would no longer be confined to the tip compartment, as previously described [24,54].

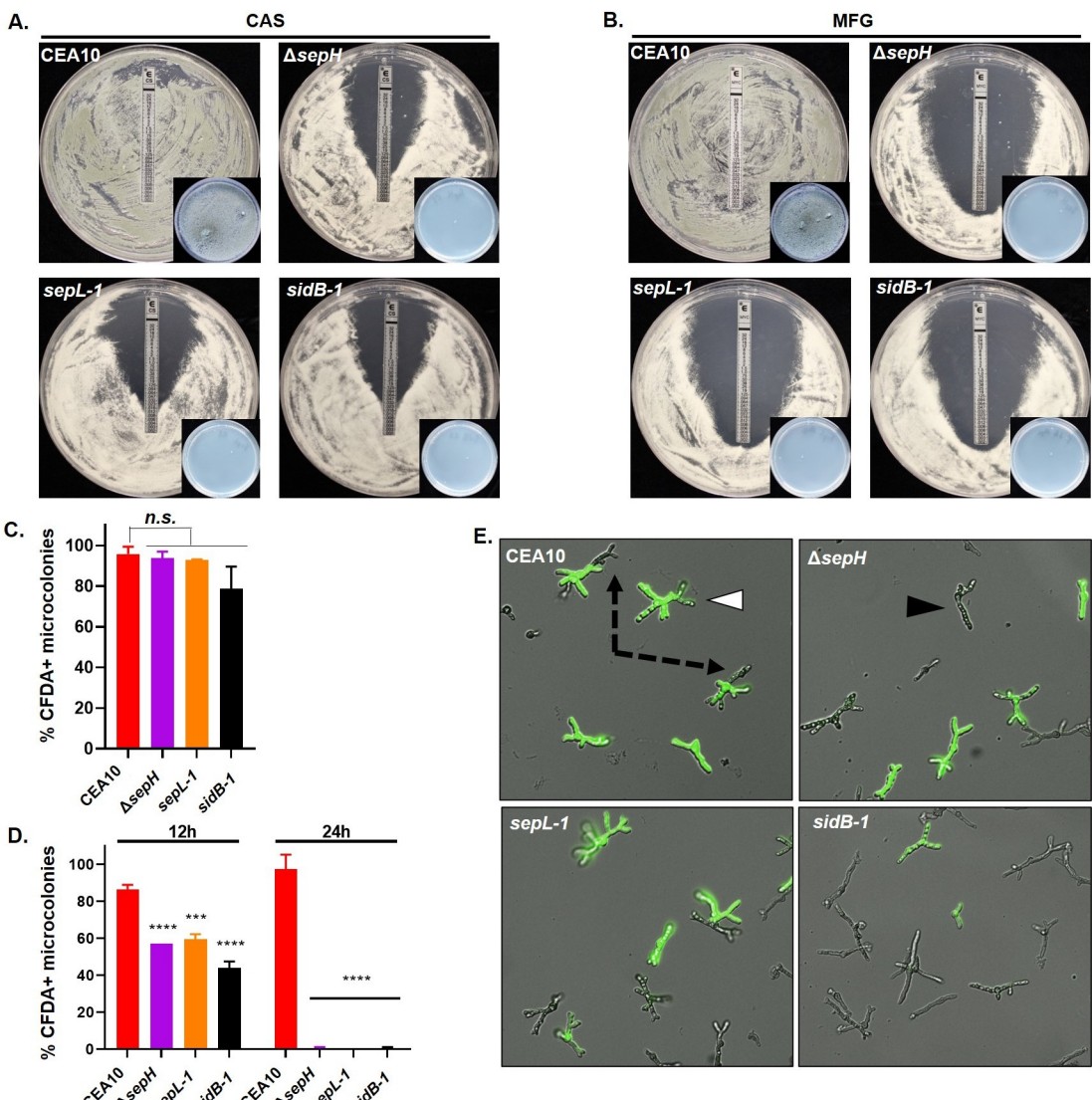

**Fig 5. The *A. fumigatus* SIN kinases are required for survival under echinocandin stress. A, B)** Loss of *sepH*, *sepL*, or *sidB* increases susceptibility to echinocandins in a modified E-test assay. 1 x 10⁶ total conidia in 500 μl sterile water from the wild type parent (CEA10), the *sepH* deletion (Δ*sepH*), or the *sepL* (*sepL-1*) or *sidB* (*sidB-1*) disruption strains were spread evenly over GMM agar plates. E-test strips for caspofungin (**A**) or micafungin (**B**) were applied and assays incubated for 48 hrs. Note the zone-of-clearance with no detectable growth for the Δ*sepH*, *sepL-1* and *sidB-1* mutants in the presence of either echinocandin. Insets show representative, drug-free minimal media culture plates onto which a single agar plug from the zone-of-clearance for each assay was sub-cultured. Multiple agar plugs (n = 10), taken from within 1 cm of the E-test strip and between the 32 and 0.25 ug/ml markers, were sub-cultured in the same manner for each assay. Note lack of growth for the SIN kinase mutant subcultures. CAS = caspofungin, MFG = micafungin. **C)** Quantitation of viability by CFDA staining of the CEA10 control and SIN kinase mutants in the absence of echinocandin stress. Conidia from each strain were germinated for 12 hrs and subsequently stained with 5-carboxyfluorescein diacetate (CFDA) to detect live hyphal elements. **D)** Quantitation of viability by CFDA staining of the strain set in the presence of caspofungin. Conidia from each strain were cultured for 12 hrs and 24 hrs at 37°C in the presence of 0.5 μg/ml caspofungin and subsequently stained with CFDA to detect live microcolonies. CFDA positivity was scored for 100 microcolonies in each experiment and all assays were completed in triplicate. Data were averaged for each strain and treatment. One-way ANOVA and Dunnett's multiple comparisons post hoc analyses indicated differences in CFDA staining in the absence of micafungin were not significant (n.s.), whereas the Δ*sepH*, *sepL-1* and *sidB-1* mutants were significantly less viable after 12 and 24 hrs growth in the presence of caspofungin. ****p<0.0001; ***p = 0.0001. **E)** Echinocandin stress during early growth stages leads to death of the SIN kinase mutants. White arrowhead denotes example of a microcolony stained positive with CFDA (bright green). Black arrowhead denotes a CFDA-negative microcolony. Dash-lined arrows denote dead (CFDA-negative) hyphal compartments of CFDA-positive microcolonies only seen in the CEA10 control.

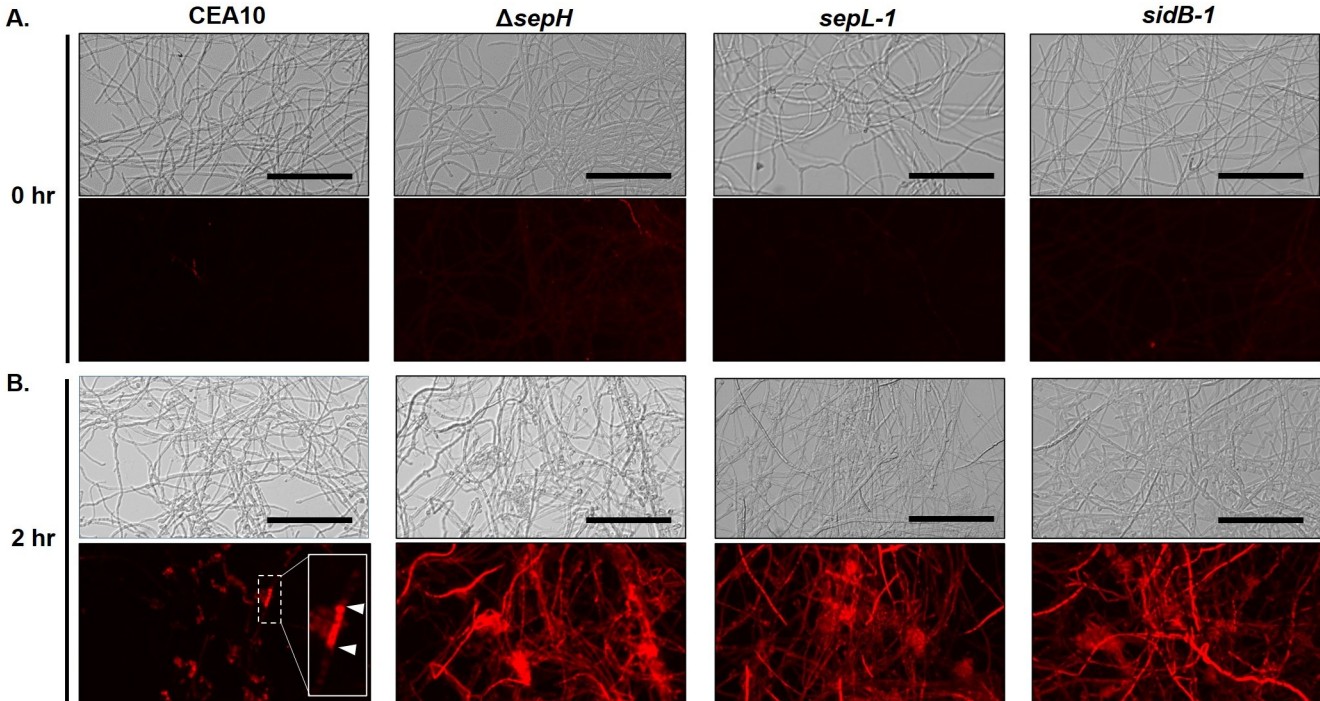

**Fig 6. Hyphae of SIN kinase mutants exhibit extensive damage in the presence of echinocandin.** Analysis of hyphal integrity using propidium iodide (PI) permeability as a measure of damage in response to echinocandin stress. Mature hyphae from the CEA10, Δ*sepH*, *sepL-1*, and *sidB-1* strains were stained with PI (12.5 µg/ml) before **(A)** or after 2 hours **(B)** exposure to micafungin (0.5 µg/ml). Upper panels are brightfield images and lower panels are fluorescence acquired. Hyphae from all strains exhibited minimal staining with no exposure to echinocandin, suggesting intact cell walls **(A)**. Limited staining of hyphal compartments was noted in the CEA10 parental strain after 2 hrs micafungin exposure, suggesting cell wall damage limited by the presence of septa (B, lower panel inset, white arrowheads denote hyphal compartment). In contrast, extensive PI staining was induced after micafungin treatment in each of the SIN kinase mutant strains **(B)**. All fluorescence images were acquired at using identical exposure. Scale bar = 100 µm.

To test this, conidia from the parental and each mutant strain were cultured to mature hyphal growth before addition of micafungin (0.5 µg/ml) and hyphal damage was subsequently analyzed by propidium iodide (PI) permeability [55]. In the absence of echinocandin-induced stress, hyphae from each strain showed little-to-no permeability to PI, supporting the CFDA staining results and suggesting that loss of SIN pathway function alone does not significantly impact cell wall integrity (Fig 6A). In contrast, hyphal damage induced by 2 hours of micafungin treatment of the SIN kinase mutants was extensive and widespread when compared to the wild type parental control (Fig 6B). This was evidenced by increased uptake of PI in each SIN kinase mutant after micafungin exposure, highlighting full hyphal elements (Fig 6B). In contrast, the CEA10 parent strain stained only minimally with PI after micafungin exposure and staining was largely restricted to specific hyphal segments (Fig 6B, inset white arrowheads). Together, these results suggested that SIN kinase activity is required for septation which, in turn, is required for limiting hyphal damage in the presence of echinocandin antifungals. Removing this barrier appeared to result in cidal anti-*Aspergillus* activity against each of the SIN kinase mutants.

## *A. fumigatus* SIN kinases are required for tissue invasive growth

To examine the impacts of hyphal septation loss on virulence, we next compared the CEA10 parent and SIN kinase mutant strains in two well-described mouse models of invasive aspergillosis, representing chemotherapeutic and corticosteroid-induced immune suppression [56].

Mice (n = 8 / arm) were immune suppressed with injections of cyclophosphamide and triamcinolone acetonide or with triamcinolone acetonide alone for the chemotherapeutic and corticosteroid model, respectively. For both models, mice were intranasally inoculated with $1 \times 10^5$ conidia and survival was followed for 15 days post-inoculation. Sham treated mice (n = 5 / arm), receiving only intranasal sterile saline inoculations coupled with immune suppressive regimens, resulted in no mortality. For the CEA10 parent strain, mortality began at Day +4 in both models with 100% mortality reached by Day +7 in the chemotherapeutic model and mortality reaching 60% by Day +15 in the corticosteroid model (Fig 7A and 7B). As a measure of virulence in a strain where SIN kinase pathway activity was restored after disruption, the *sepL-1-C'* complement strain induced mortality statistically similar to that of the CEA10 parent in both models (Fig 7A and 7B). Surprisingly, the SIN kinase mutants were avirulent in both models, with the only exception being the *sepL-1* mutant which induced 50% mortality only in the chemotherapeutic model (Fig 7A and 7B). All SIN kinase mutant-induced mortality levels were significantly reduced, compared to the parent strain. To examine the histopathological impact of hyphal septation loss on infection, we also analyzed hyphal growth *in vivo* through silver-stained tissue sections of infected lungs from each group. At 4 days post-infection, the wild type CEA10 strain had generated large, deeply invasive hyphae (Fig 8). In contrast, each of the SIN kinase mutants had formed small hyphal masses residing only in the open airways at the same timepoint post-infection (Fig 8). No deeply invasive growth was noted for any SIN kinase mutant. A single instance of shallow invasion was noted for the Δ*sepH* mutant and this was associated with what appeared to be a loss-of-polarity phenotype characterized by highly branched hyphal tips (Fig 8, inset). These data suggested that the SIN kinase pathway is essential to virulence through support of invasive tissue growth.

To quantitatively analyze SIN kinase mutant fitness and host-pathogen interaction, we next completed qPCR-based fungal burden assays, as previously described [57]. Mice (n = 5 / group) were immune suppressed following the chemotherapeutic protocol and subsequently infected with $1 \times 10^6$ conidia from each strain by intranasal inoculation. At Day +4, lungs were aseptically removed and processed for genomic DNA extraction. qPCR-based quantitation of *A. fumigatus* DNA in lung tissues revealed that both the Δ*sepH* and *sidB-1* mutants accumulated significantly less fungal burden *in vivo* than the CEA10 parent (Fig 9A). Although the *sepL-1* mutant displayed reduced burden by qPCR when compared to CEA10, this difference was not statistically significant (Fig 9A). This inability to accumulate wild type levels of fungal mass *in*

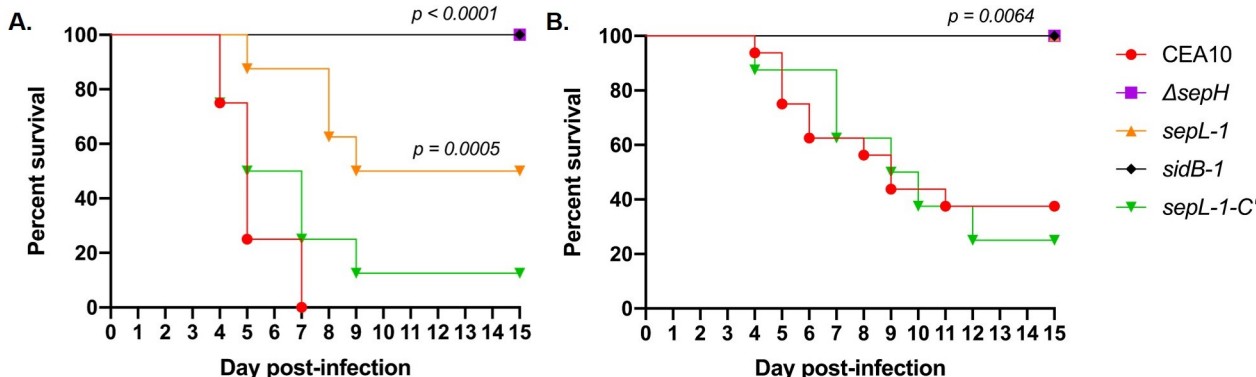

**Fig 7. SIN kinase activity is required for virulence in mouse models of invasive aspergillosis.** Mice (n = 8 / group for Δ*sepH*, *sepL-1*, *sidB-1* and *sepL-1-C'*; n = 16 for CEA10) were chemotherapeutically immune suppressed with both cyclophosphamide and triamcinolone acetonide (**A**) or triamcinolone acetonide alone (**B**) and inoculated with $1 \times 10^5$ conidia of the indicated strain. Survival was followed for 15 days post-inoculation. Statistical analyses (Mantel-Cox Log-rank test) identified significantly reduced virulence for all SIN kinase mutant strains *vs.* the CEA10 control.

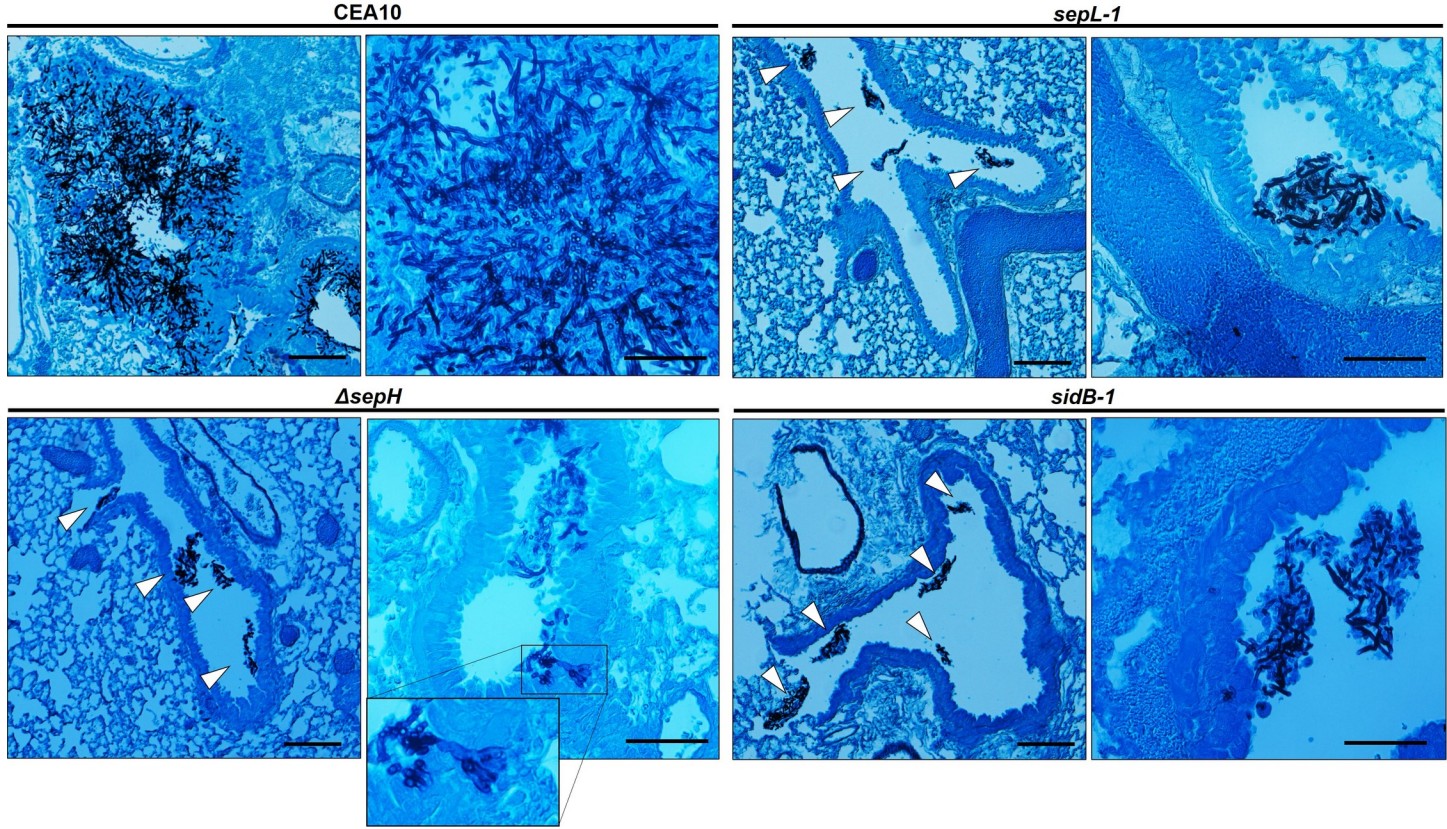

**Fig 8. Loss of virulence among the SIN kinase mutants is associated with lack of tissue invasion.** Low- and high-magnification photomicrographs of Gomori methenamine silver (GMS)-stained lung tissue sections from the CEA10, Δ*sepH*, *sepL-1* and *sidB-1* at day +4 post-inoculation. Mice were immune suppressed with triamcinolone acetonide and inoculated with each strain as described for the previous survival studies. Hyphae (black stained fungal elements) from the CEA10 strain were noted to invade lung tissue, forming fulminant lesions. In contrast, growth of each SIN kinase mutant was limited to the airways with minimal to no tissue invasion (white arrowheads). Rare tissue invasion was associated with loss of polarity maintenance (Δ*sepH* inset panel). Scale bar = 50 μm.

*vivo* was also associated with a diminished ability to induce pro-inflammatory cytokine release *in vivo*. ELISA-based detection of IL-1β and TNFα in lung homogenates from the same mice utilized for fungal burden revealed that all SIN kinase mutants induced significantly lower cytokine levels compared to the CEA10 control (Fig 9B and 9C). To test if these findings were simply due to negative impacts on fitness, we compared the abilities of the SIN kinase mutants to induce IL-1β release *in vitro* using the THP-1 macrophage-like cell line. Release of IL-1β is a pro-inflammatory response typically induced by *A. fumigatus* cell wall PAMP exposure and is known to be dependent on inflammasome signaling [58,59]. Differentiated THP-1 cells were co-incubated with conidia from the wild type CEA10 or SIN kinase mutants (MOI 10:1) for 16 h and supernatants were subsequently analyzed for IL-1β release by ELISA. Surprisingly, all SIN kinase mutants were found to induce significantly lower levels of IL-1β release when compared to CEA10 (Fig 9D). In addition, IL-1β release in our assay was confirmed to be dependent on activation of the NLRP3 inflammasome, as the CEA10 parental strain was unable to induce IL-1β release in NLRP3-/- or ASC-/- cells, components necessary for canonical inflammasome activation and assembly (Fig 9E) [60]. Inflammasome-dependence was further evidenced by the ability to block *Aspergillus*-induced IL-1β release using the well-characterized NLRP3 inhibitor, MCC950 (Fig 9F). Together, these data indicate that the SIN kinase pathway is not only important for supporting *A. fumigatus* pathogenic fitness but is also required for normal damage-induced immune activation.

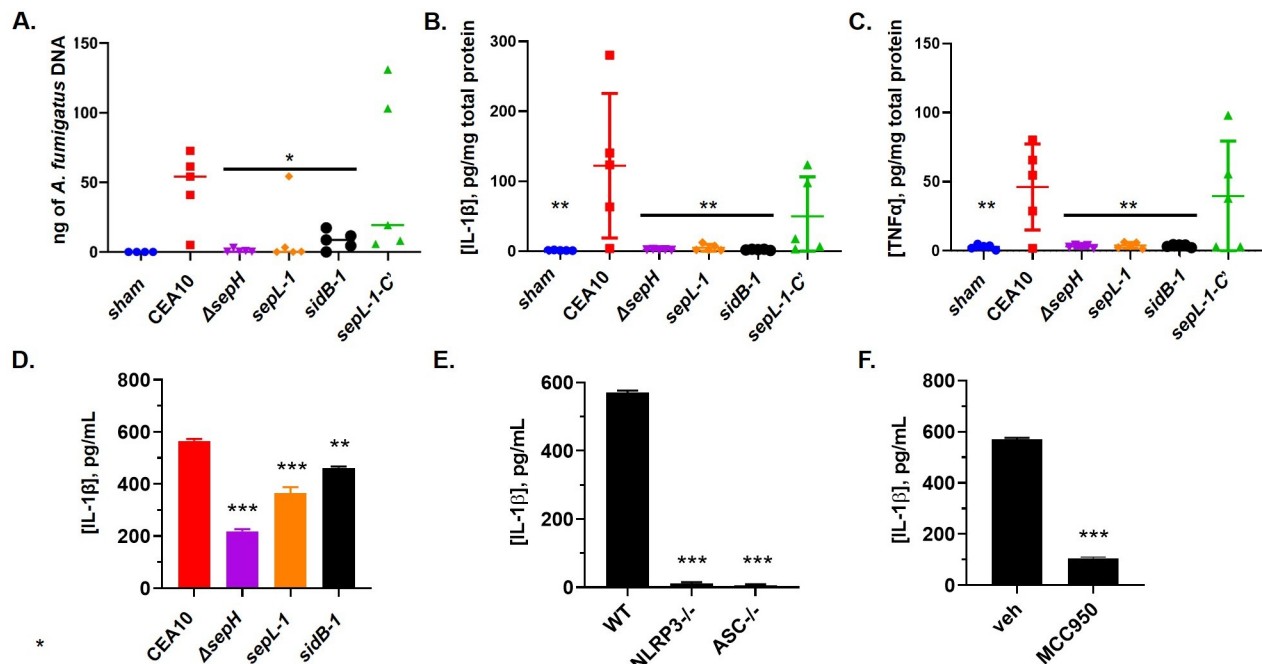

**Fig 9. Loss of virulence in the SIN kinase mutants is characterized by decreased fungal burden and host response to infection. A)** Analysis of fungal burden by qPCR at day +4 post inoculation. Mice (n = 5 / group) were immune suppressed with cyclophosphamide and triamcinolone acetonide and inoculated with 1x 10⁶ conidia from each strain. Data are represented as nanograms of *A. fumigatus* specific DNA in 500 ng of total DNA. * p < 0.02. Quantitation of IL-1β **(B)** and TNFα **(C)** revealed decreased host response in SIN kinase mutant infected mice. Mice (n = 5 / group) were immune suppressed as indicated for fungal burden analysis and lung tissue was removed at day +4 post-inoculation, homogenized and analyzed by ELISA. **p = 0.0024 for (B); **p = 0.0031 for (C). An *in vitro* IL-1β release assay uncovered decreased induction of inflammasome activation by the SIN kinase mutants. **D)** Conidia from each strain were co-incubated with phorbol 12-myristate 13-acetate (PMA)-activated THP-1 cells (MOI 10:1) for 16 hrs and supernatants analyzed by ELISA for IL-1β concentration. ***p < 0.0001; **p = 0.0014. **E)** Inflammasome dependence of IL-1β release was established by co-culturing PMA-activated WT (THP1-null), *Nlrp3⁻/⁻* (THP1-KO-NLRP3), and *Asc⁻/⁻* (THP1-KO-ASC) THP-1 cells with CEA10 conidia (MOI 10:1) as indicated for (D). ***p < 0.0001. **F)** Inflammasome dependence of *Aspergillus*-induced IL-1β release was further confirmed by repeating this assay in the presence of the inflammasome inhibitor, MCC950 (10 μM). ***p < 0.0001. All experiments were conducted in technical replicates (*n* = 4) and repeated independently in triplicate. Statistical comparisons in (A), (B), (C), and (D) were made by one-way ANOVA with Dunnett's multiple comparisons test post hoc and represent comparison of each SIN kinase mutant to the CEA10 control. Statistical comparisons in (E) were made by one-way ANOVA with Dunnett's multiple comparisons post hoc and represent the *NLRP3-/-* and *ASC-/-* versus WT control. The statistical comparison of MCC950 versus vehicle in (F) was made by unpaired T-test.

## Loss of hyphal septation improves echinocandin-mediated fungal clearance during invasive disease

Taken together, our *in vitro* and *in vivo* data suggest that *A. fumigatus* mutants lacking septa should be more susceptible to echinocandin therapy during infection. To examine the *in vivo* therapeutic relevance of our *in vitro* findings, we next employed the wild type CEA10 parent and the *sepL-1* strain in a chemotherapeutic mouse model of invasive aspergillosis with and without echinocandin therapy. The *sepL-1* mutant and chemotherapeutic model were chosen here as this combination was found to result in measurable mortality in our previous experiments. Ten mice per experimental arm were infected with 10⁶ conidia of either the CEA10 or *sepL-1* strain by intranasal inoculation on day 0. On Days +1, +2 and +3, mice were administered (or not) micafungin therapy (1 mg/kg/day or 2 mg/kg/day) by intraperitoneal injection (three total doses). Survival at the end of the 14-day infection was 0% for the CEA10 untreated arm and 75% for the *sepL-1* untreated arm (p = 0.002) (Fig 9A). Strikingly, with micafungin therapy at 2 mg/kg/day, the *sepL-1* mutant-infected mice exhibited 100% survival whereas the wild type strain-infected mice, treated in the same manner, produced 0% survival by day +9

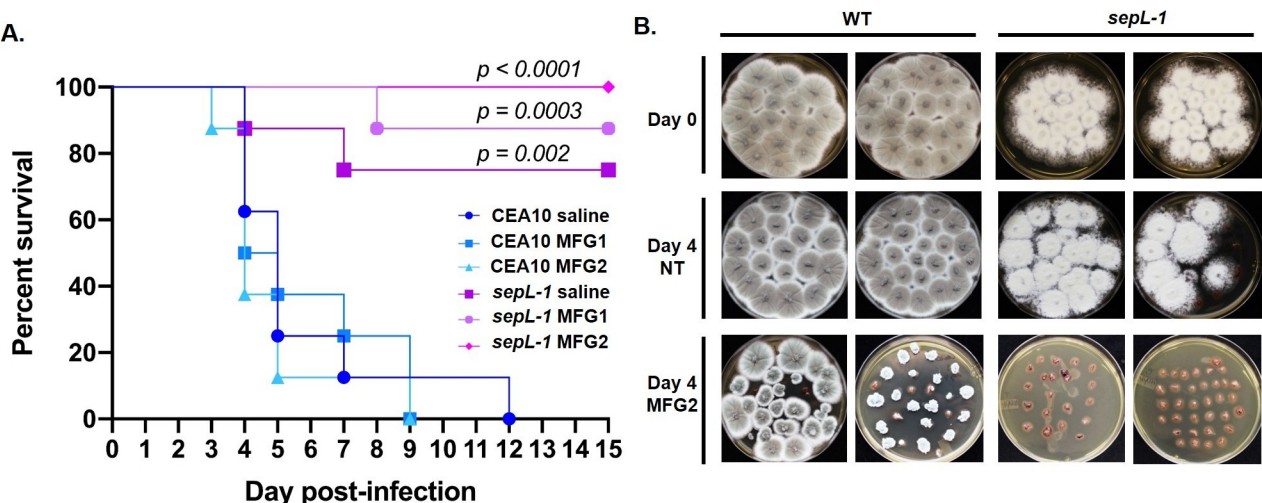

**Fig 10. Loss of hyphal septation improves echinocandin therapy characterized by clearance of fungal burden from lung tissue. A)** Survival analysis of mice infected with either the CEA10 or *sepL-1* mutant strain with and without micafungin therapy. All mice were immune suppressed through intraperitoneal injection of cyclophosphamide on days -3, +1, +4, and +7 and a single subcutaneous injection of triamcinolone acetonide on day -1. Mice were inoculated with 1 X 10$^6$ conidia of the indicated strain suspended in 20 μl of sterile saline on day 0 and then received three separate intraperitoneal injections of micafungin at either 1 mg/kg (MFG1) or 2 mg/kg (MFG2) on days +1, +2 and +3. Statistical comparisons were made by Mantel-Cox log-rank test and represent each *sepL-1* mutant experimental arm compared to its CEA10 control (i.e., CEA10 saline *vs. sepL-1* saline, CEA10 MFG1 *vs. sepL-1* MFG1, and CEA10 MFG2 *vs. sepL-1* MFG2). **B)** CEA10 and *sepL-1* residual lung tissue burden at day 0 and day 4 with and without micafungin 2 mg/kg therapy. Organ cultures are shown from two representative animals from each treatment group. Note that the *sepL-1* infected mice treated with micafungin 2 mg/kg are culture negative at day 4 post-inoculation. MFG = micafungin.

(p < 0.0001) (Fig 9A). Although survival differences between treated and untreated CEA10 and *sepL-1* strains were significant for each therapeutic regimen, we found no significant difference between *sepL-1* treated and untreated experimental arms (Fig 10A). This was due to a combination of the low virulence of the *sepL-1* mutant and the number of mice used per arm. However, in a separate experiment, the ability of micafungin therapy to enhance the reduction of residual tissue burden in *sepL-1* infected mice was also examined. Mice (n = 8 / arm) were immune suppressed and inoculated identical to survival assays and were provided (or not) micafungin therapy at days +1, 2, and 3. Lungs were removed at day +0 or day +4, sectioned, and cultured on Inhibitory Mold Agar at 37°C for 48 hours. All non-micafungin treated mice infected with either the CEA10 or *sepL-1* strains produced positive fungal cultures using tissue extracted at day +0 and day +4, indicating live fungus in the lung environment for both strains at this timepoint (Fig 10B). At day +4 with micafungin therapy (2 mg/kg/day), 75% of CEA10-infected mouse lung cultures (6 / 8) were still positive for fungal growth (Fig 10B). Compared to fungal colony morphologies arising from micafungin-free lungs (CEA10 day +0 or day +4, no micafungin), the micafungin-treated mouse lung cultures produced compact colony growth confirming presence of micafungin in tissues during therapy (Fig 10B). In contrast, lungs from *sepL-1* infected, micafungin-treated mice extracted at day +4 contained no culturable fungal elements (Fig 10B). Together, these data suggest that loss of hyphal septation improves echinocandin therapy by enhancing the ability of this drug class to clear invading *A. fumigatus* hyphae from the lung.

## Gene deletion of additional septation mediators phenocopies the SIN kinase mutants

As the striking loss of virulence and tissue invasion phenotypes of the SIN kinase mutants could be dependent or independent of hyphal septation, we next wanted to test whether these

phenotypes could be replicated in a SIN kinase-independent but septation-dependent manner. Therefore, we sought to identify novel genes essential for septation in *A. fumigatus*. In the model filamentous fungus, *A. nidulans*, the *acnA* gene encodes an alpha-actinin protein that is essential for hyphal septation and loss of septation in a strain lacking *acnA* is associated with a complete absence of contractile actin ring assembly [61]. This data mirrors results reported for a *Schizosaccharomyces pombe* alpha-actinin gene, *ain1*, which is also mediates CAR assembly in the model fission yeast [62]. Additionally, in *Saccharomyces cerevisiae*, the *MLC1* gene encodes for a myosin light chain protein that is essential for regulating myosin heavy chain interactions during contractile actin ring assembly [63]. Loss of *S. cerevisiae MLC1*, or the *mlc1* ortholog in *S. pombe* (cdc4), results in lethality for both yeasts due to the inability to complete CAR formation and subsequent cytokinesis [64,65]. Given their conserved roles in septation and the non-essential nature of septation in *A. fumigatus*, we reasoned that we should be able to acquire gene deletions of the *A. fumigatus* orthologs of *acnA* and *MLC1* for further study here. The putative protein sequences for *acnA* and *MLC1* from *A. nidulans* and *S. cerevisiae*, respectively, were utilized for a BLASTp search of the *A. fumigatus* genome (FungiDB). For *MLC1*, this search identified two proteins with significant identity to the Mlc1p sequence. These were proteins encoded by the uncharacterized gene, AFUB_091530 (42% identity), and an ortholog of the highly conserved calmodulin gene, AFUB_067160 (34% identity). Therefore, AFUB_091530 was chosen as the *A. fumigatus MLC1* ortholog and named *mlcA*. Surprisingly, no orthologs of the *A. nidulans* AcnA protein could be identified in *A. fumigatus*. However, additional BLASTp searches using the *S. pombe* Ain1 protein sequence identified a single *A. fumigatus* gene, AFUB_055850, with 49% identity to Ain1 and was therefore named *ainA*. This BLASTp search also identified a single Ain1 ortholog in *A. nidulans* (AN7707) that was unique from the aforementioned AcnA protein. Employing CRISPR/Cas9-based gene targeting, single gene deletion and complementation mutants were generated for both genes (S3 Fig).

As was observed for the SIN kinase mutants, CFW staining of mature hyphae revealed loss of septum formation in the Δ*mlcA* and Δ*ainA* mutants. Whereas the parental wild type CEA10 displayed fully formed septa after 20 hours of culture in minimal media, the Δ*mlcA* and Δ*ainA* mutants formed completely aseptate hyphae. The Δ*mlcA* mutant was additionally characterized by the presence of brightly stained CFW-positive puncta of cell wall material throughout hyphae (Fig 11A). This finding suggests that, in addition to loss of septum formation, Δ*mlcA* deletion causes abnormal deposition of cell wall material during hyphal growth. E-test assays, employing anidulafungin-embedded strips, resulted in the formation of a zone-of-clearance for both the Δ*mlcA* and Δ*ainA* mutants (Fig 11B). These results were similar to those generated by the SIN kinase mutants (Fig 5A and 5B). These findings were, again, in contrast to the wild type CEA10 parent that formed only a zone of depressed growth in response to echinocandin stress by E-test (Fig 11B). When employed for survival analyses in a corticosteroid model of invasive aspergillosis, Δ*mlcA* and Δ*ainA* again phenocopied the SIN kinase mutants. At 15 days post-inoculation, both mutants resulted in no mortality whereas the wild type and complement strains performed similarly and produced significantly higher mortality (Fig 12A and 12C). Further, histopathological analysis of silver-stained tissue sections from lungs of mlcA and ainA infected mice revealed growth of fungal elements only in the airways (Fig 11B and 11D). These results again mirrored those for the SIN kinase mutants where a lack of tissue invasion was noted for aseptate hyphae (Fig 8). Together, these findings suggest that deletion or disruption of the signaling pathways or machinery required for septum initiation and formation results in the inability to support invasive growth in the host lung environment.

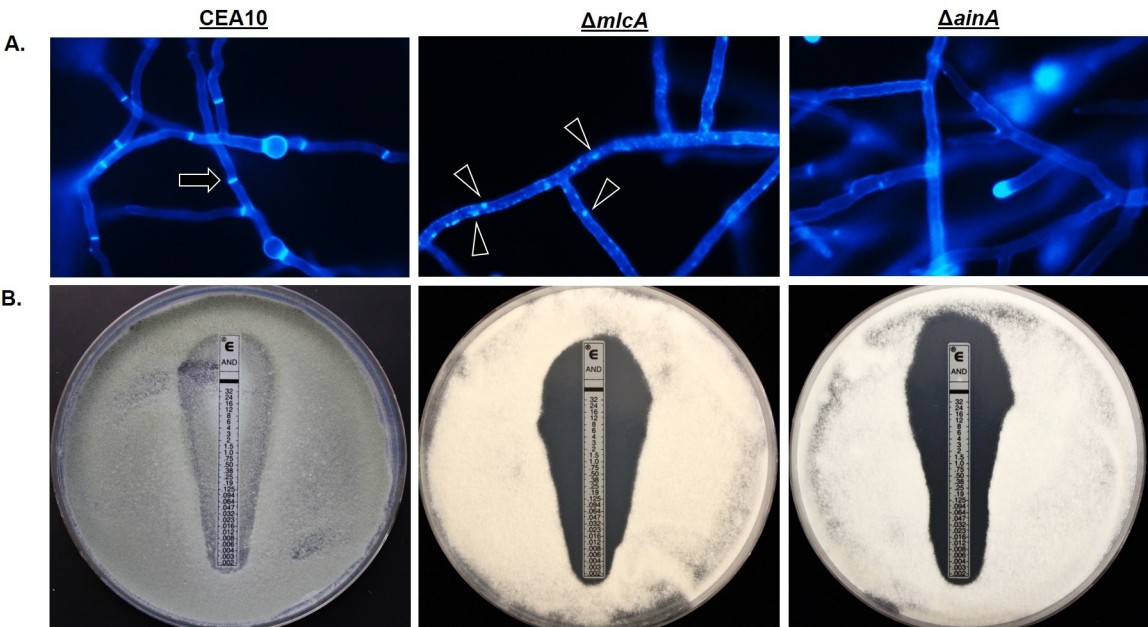

**Fig 11. The *A. fumigatus* genes encoding myosin light chain (*mlcA*) and alpha-actinin (*ainA*) are required for hyphal septation and echinocandin resistance. A)** Characterization of septation by CFW staining of the wild type parental strain (CEA10) and the *mlcA* (Δ*mlcA*) and *ainA* (Δ*ainA*) deletion mutants. Conidia from each strain were cultured to mature mycelial growth on sterile coverslips submerged in minimal media. Microscopic analysis of CFW-stained cultures revealed fully formed, normal septa in the CEA10 control strain (black arrow), whereas Δ*mlcA* formed aseptate hyphae with brightly stained puncta of cell wall material and Δ*ainA* developed only aseptate hyphae. **B)** Anidulafungin E-test assays for each strain. Note the zone-of-clearance of the Δ*mlcA* and Δ*ainA* strains which show the complete absence of growth.

## Discussion

Hyphal septa are the product of incomplete cytokinesis and have long been appreciated to function as a protective barrier to mechanical and chemical stresses that disrupt the cell walls of filamentous fungi. This was the first investigation of the function of the SIN kinases in the human pathogen, *A. fumigatus*. The SIN complex has been studied most intensely in the fission yeast, *Schizosaccharomyces pombe* [66]. Although the SIN-complex shares similarity and encompasses some protein components homologous to those comprising the Mitotic Exit Network (MEN) in other yeast species like *Saccharomyces cerevisiae*, the signaling pathways are distinct [37,38]. The core of the SIN in *S. pombe* is composed of a GTPase protein, Spg1, that is negatively regulated by a two-component GTPase activating protein, Cdc16/Byr4 [37]. Although biochemical evidence is lacking even in the model organisms, this module is thought to signal through a kinase cascade consisting of the Cdc7 –Sid1 –Sid2 protein kinases with the co-regulatory components, Cdc14 and Mob1, binding to and regulating the Sid1 and Sid2 kinases, respectively [37]. The core tripartite kinase cascade is conserved in the model filamentous fungi *A. nidulans* (SepH-SepL-SidB) and *Neurospora crassa* (CDC-7, SID-1, DBF-2) [39,40] and the orthologs of these kinases were the focus of this study in *A. fumigatus*. Although downstream SIN-complex effectors are not verified in *A. fumigatus*, the Rho-type GTPase, Rho4, and the putative Rho Guanine Exchange Factors, Bud3 and Rgf3, are known to regulate septum formation and a *rho4* deletion is the only mutation in *A. fumigatus* previously reported to completely block septation [52,54,67]. Rho4 has been proposed to be a downstream effector of SIN-kinase activity that, in turn, recruits a formin protein (SepA ortholog in *A. fumigatus*) to mediate assembly of the contractile actomyosin ring (CAR) on which the

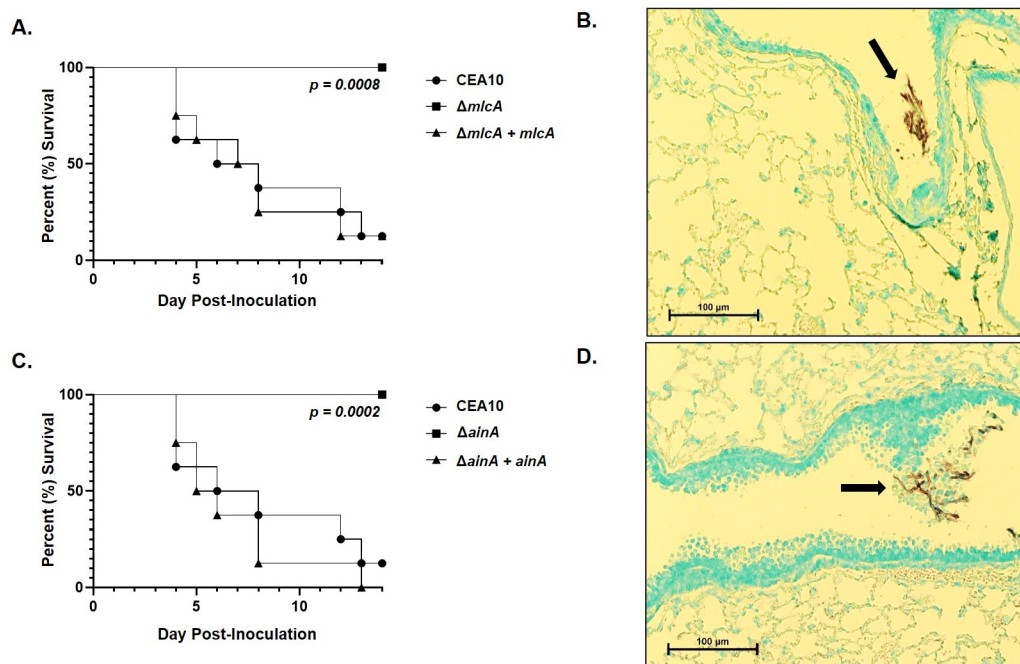

**Fig 12. Loss of septation caused by deletion of *mlcA* or *ainA* results in avirulence associated with lack of tissue invasion.** Survival analysis and GMS-stained tissue histology from immune suppressed mice infected with the Δ*mlcA* isogenic strain set **(A)** and **(B)** or the Δ*ainA* isogenic strain set **(C)** and **(D)**, respectively. Mice (n = 8 / strain) were immune suppressed with a single, subcutaneous injection of triamcinolone acetonide (40 mg/kg) on Day -1. On Day 0, all mice were intranasally administered 100,000 conidia suspended in 20 μl of sterile saline. No deaths were recorded in sham treated mice (sterile saline alone, n = 8). Statistical comparisons were made by Mantel-Cox log-rank test. Black arrows denote non-invasive fungal growth identified only in the airways of Δ*mlcA* and Δ*ainA* infected mice.

septum is built in *A. nidulans* [68]. The CAR is further built and constriction regulated by bundling of actin filaments through the action of α-actinin cross-linking proteins and by interactions with myosin light and heavy chain proteins [69]. Interestingly, although the *A. nidulans* α-actinin, AcnA, has been shown to regulate septation, we found this gene to not be conserved in *A. fumigatus*. This finding suggests differences in how *A. fumigatus* and *A. nidulans* regulate septum formation. In contrast, we found that the α-actinin gene, *ainA*, as well as the myosin light chain gene, *mlcA*, were conserved among *A. fumigatus*, *A. nidulans* and *S. pombe*. Both genes played essential roles in completion of septation. This event is intimately coupled with the exit from mitosis in *S. pombe* such that SIN-complex mutants are typically inviable in this yeast. In filamentous organisms like the *Aspergilli*, however, mitosis and septation are not interdependent and, as our data show, aseptate *Aspergillus* mutants are viable under normal growth conditions.

The initial publications describing anti-*Aspergillus* echinocandin activity as being characterized by fungal hyphal tip lysis, were also the first to suggest the possibility that hyphal septation likely underpins the fungistatic nature of this drug class against *Aspergilli* [24]. This conclusion was further supported by subsequently published genetic and pharmacologic evidence. For example, when the echinocandin-target gene *fksA* is deleted in *A. fumigatus*, hyphae are devoid of β-1,3-glucan and appear to be killed in the presence of compounds that inhibit septation [54]. Additionally, the aseptate Δ*rho4 A. fumigatus* mutant cannot grow in the presence of echinocandins [52]. Although highly suggestive of an essential role for hyphal septation in protection against echinocandins, the septation inhibitors previously employed,

hydroxyurea and diepoxyoctane, potentially have many off-target effects as developmental and cell cycle inhibitors and the *rho4* gene was additionally characterized as directly contributing to modulation of the cell wall [52]. Our data definitively show that loss of *A. fumigatus* hyphal septation imparts fungicidal activity to echinocandins. Thus, septation is a mechanism to limit damage imposed by loss of cell wall integrity via inhibition of β-1,3-glucan synthesis. When faced with echinocandin stress, the FksA enzyme mislocalizes from its normal position at the hyphal tip and the tip compartment is lysed [70]. However, the interseptal compartments remain viable [24,70]. In many septation-competent fungi, septal pores that remain after septum formation to promote exchange of cytoplasmic materials plug quickly after cell wall damage to protect against cytoplasmic leakage [71,72]. After tip-lysis by echinocandins, new hyphal growth within the lysed hyphal tip compartments, known as "intrahyphal hyphae", can develop from existing septa in the proximal subapical compartment [70]. The FksA enzyme is relocalized to these new sites of apical growth to support the nascent growth axis. Therefore, the septum may act not only as physical barrier to limit hyphal damage to the colony periphery, but also as a scaffold for continued hyphal growth under cell wall stress in the *Aspergilli*.

Under the conditions we tested, the aseptate SIN kinase mutants were more susceptible to echinocandin-mediated cell wall damage than even the CWI kinase mutants, which represent the core signaling pathway of cell wall stress responses (S4 Fig). The majority of published works identifying cellular mechanisms supporting *A. fumigatus* survival in response to echinocandins have logically focused on adaptive cell wall stress signaling, such as the CWI kinase cascade and cAMP-dependent protein kinase A signaling [21,32,34]. Although they represent potentially powerful targets for future combination therapies with echinocandins, inhibiting many of these pathways leaves the cell septation machinery largely intact and, therefore, provides the pathogen a means for persistence in the host under echinocandin stress. Recent studies suggest that the process of septation may also be stress responsive in fungi. In the yeast pathogen *Candida albicans*, loss of the *CHS1* chitin synthase gene is lethal due to the central role of the Chs1p enzyme in building the primary septum for successful cell division [73]. However, induction of cell wall stress in a *CHS1* repressed strain causes stress-induced formation of an alternative "salvage septum" by other chitin synthase enzymes, leading to completion of cytokinesis and cell survival in the absence of *CHS1* [74]. Therefore, the enzymes that are typically required for the physical construction of the septum can be re-wired under cell wall stress to promote survival in *C. albicans*. In *A. nidulans*, a recent phosphoproteomics analysis of micafungin-stressed hyphae identified the SIN network ortholog, SidB, as being hyperphosphorylated in a manner likely dependent on the CWI MAPK, MpkA [30]. Further *in vitro* growth analyses analyzing septum formation in micafungin-stressed and micafungin-free samples revealed that the rate of septation increases during echinocandin-induced cell wall stress [30]. Together, these data argue that formation of this important protective barrier in filamentous fungi may be both a fundamental structural cellular component and stress responsive.

A plethora of previous studies have shown that neutrophil activity is required for clearance of hyphae and hyphal fragments during infection [75]. Patients most at-risk for invasive aspergillosis are profoundly neutropenic and, as such, viable hyphal fragments remaining after unsuccessful therapy or under sub-therapeutic levels of drug are potentially especially problematic. Our *in vivo* virulence data suggests that blockade of septation could negatively impact *A. fumigatus* persistence by promoting echinocandin-induced hyphal death, likely before effective adaptive responses can be upregulated. The nearly complete lack of virulence of the SIN kinase mutant strain was a surprising finding and is the first description of a connection between hyphal septation and virulence in a filamentous human pathogen. The molecular and cellular mechanisms for why septation is required for *A. fumigatus* virulence are unknown.

However, loss of hyphal septation has been shown to block virulence of the smut fungus, *Ustilago maydis*, due to loss of needed turgor pressure to build the appressorium [76]. Although human infectious fungi do not utilize special infection structures like appressoria for invasion, the inability of *A. fumigatus* aseptate hyphae to invade murine lung tissue could be from lack of turgor pressure in the tip compartment required for physical invasion. A recent study exploring the connections between hyphal growth and polarity maintenance has found filamentous fungal organisms with fast growing hyphal tips and high turgor pressure to often lose polarity when undergoing invasive growth (i.e., penetration into small spaces) [77]. in contrast, slow growing fungi with lower turgor pressure are able to maintain a single polarized growth axes under the same conditions. Therefore, a trade-off exists between growth rate and morphological plasticity for these fungi that is especially important during invasive growth into substrates. As we noted loss of polarity in our SIN kinase mutants during attempted tissue invasive growth (Fig 8), it may be possible that septation is involved in imposing such a trade-off between growth rate, turgor pressure and maintenance of hyphal morphogenesis for *A. fumigatus*. Although we utilize multiple septation-deficient mutants in our study, our data do not strictly rule out the possibility that hypersusceptibility of the SIN kinase mutants to cell wall stress could be at least partially septum-independent. For example, the hyphal septation machinery may be required to maintain cell wall stability or may crosstalk with cell wall biosynthesis pathways for support of overall hyphal structure during tissue invasive growth. This explanation would suggest a regulatory link between septation and cell wall integrity / biosynthesis networks. In support of this possibility, our finding that the SIN kinase mutants displayed reduced stimulation of IL-1β release *in vitro* may argue that cell wall PAMP exposure and / or cell wall structure are impacted by loss of septation. Further in-depth studies on SIN kinase signaling and cell wall integrity pathways will fully delineate the septation-dependent and -independent mechanisms underpinning the tissue invasion phenotypes resorted here.

In conclusion, we report for the first time that hyphal septation in *A. fumigatus* is required for virulence of this important human pathogen. We also show, definitively, that loss of hyphal septation generates cidal activity of echinocandins against *A. fumigatus* and enhances *in vivo* echinocandin activity by promoting clearance of viable residual tissue burden. Together, our findings suggest that inhibitors of septation could enhance echinocandin-mediated killing while simultaneously limiting the invasive potential of *A. fumigatus* hyphae.

## Materials and methods

### Ethics statement

All experiments involving animals were approved by the Institutional Animal Care and Use Committee at the University of Tennessee Health Science Center under protocol number 19–0067.

### *A. fumigatus* strains and growth conditions

The wild type strain, CEA10, was utilized as the parental strain for all genetic manipulations described herein. All strains generated as part of this study are listed in S1 File. For quantification of colony diameter, five microliters of sterile water containing $10^4$ conidia were inoculated onto the center of glucose minimal media (GMM) agar plates [78] and incubated at 37°C. Colony diameters were measured every 24 hours and pictures were taken after 96 hours incubation. Colony diameters of each mutant were compared to those of the parental strain, CEA10, using two-way ANOVA with Tukey's test for multiple comparisons (GraphPad Prism v8.2.1). To evaluate and quantify the production of conidia, $2\times10^4$ conidia of each mutant were inoculated onto the center of GMM agar plates and incubated for 4 days at 37°C. After this time,

colony diameters were measured and conidia were harvested in identical volumes of sterile water from each plate. Recovered conidia were counted using a hemocytometer and results were expressed as conidia per mm$^2$ of colony area. Results from the kinase disruption mutants were compared to CEA10 using one-way ANOVA followed by Dunnett's test for multiple comparisons (GraphPad v8.2.1). Conidial quantification and colony diameters were determined at least twice for each mutant.

### Genetic manipulations of *Aspergillus fumigatus*

All putative protein kinase encoding genes were identified via BLAST search of the A. fumigatus genome database (FungiDB.org) using the known protein kinases of *Aspergillus nidulans* [39]. In total, 142 putative protein kinase encoding genes were identified and are listed in S1 File. Each putative protein kinase gene was targeted for disruption in a wild type genetic background (CEA10) using a CRISPR-Cas9 gene editing technique previously described by our laboratory [41]. To increase throughput of mutant generation, standard *A. fumigatus* protoplast-mediated transformation protocols were adapted to a miniaturized, 96-well plate system.

Briefly, CRISPR RNAs (crRNAs) and primers used to amplify hygromycin resistance cassette repair templates, engineered to incorporate 40 bp micro-homology regions at both 5' and 3' ends, were designed using the Eukaryotic Pathogen CRISPR guide RNA/DNA Design Tool (EuPaGDT) from the University of Georgia (http://grna.ctegd.uga.edu/). This tool allows users to identify Protospacer Adjacent Motifs (PAM) sites and protospacers in desired regions of the genome while predicting off-target sites. Protospacer regions were selected within the first exon of each gene, when possible. Repair templates were designed to delete 5 nucleotides of the Open Reading Frame (ORF) upon integration in an attempt to generate a frameshift and, consequently, disrupt gene function in the event of read-through during transcription. Forward and reverse primers for generation of repair templates were purchased in arrayed 96-well plates and utilized for PCR reactions also carried out in 96-well plates. Five microliters of each PCR reaction was utilized for gel electrophoresis to ensure a single band of appropriate size was generated for each reaction. After confirmation of successful PCR, five microliters of the unpurified PCR reaction, containing about 200 ng of amplified repair template, was utilized for transformation. All primers utilized for repair template generation are listed in S1 File.

Guide RNAs (gRNAs) and Ribonucleoprotein (RNP) complexes for each gene were built *in vitro* using commercially available tracrRNA and Cas9 enzyme, as previously described [41]. To reduce the cost of each transformation, as well as the time required for library construction, we adapted a traditional *A. fumigatus* protoplast transformation procedure to a miniaturized system, which allowed the performance of each gene disruption in a final total volume of 200 μl. The transformations were carried out in 96-well plates (Fig 1), which allowed 96 different transformations in a single day. Briefly, 96-well plates with each well containing a transformation mixture composed of protoplasts (1–5 x 10$^5$), Cas9 RNP complex (1.2 μM gRNA + 500 ng Cas9), repair template (~200 ng), 60% polyethylene glycol 3350 (PEG 3350), and 11 μl of STC Buffer [79] were incubated on ice for 50 min. After this time, 57 μl of 60% PEG were added to each well and incubated at RT for 20 min. Finally, the volume of each well was brought to 200 μl using STC buffer and the entire contents of each well were plated onto a single Sorbitol Minimal Medium [79] agar plate. The protoplasts were allowed to recover by incubating the culture plates at RT overnight, and top agar (SMM with 0.75% agar) containing 450 μg / ml of hygromycin was added the next day. The plates were then incubated at 37˚C until colonies were observed. After 3–4 days, single colonies were transferred to new GMM plates supplemented with 150 μg / ml of hygromycin, single spored and genotypically screened by multiple PCRs, as seen in Fig 1D.

To generate *sepH*, *mlcA*, and *ainA* deletion strains, CRISPR/Cas9-mediated gene targeting was employed, as previously described [41]. To aid in complete gene deletion, two PAM sites, located upstream and downstream of the respective genes, were selected and HygR repair templates were designed to contain microhomology regions targeting areas outside each PAM. Transformation was carried out as described above and positive transformants were screened by PCR. Complementation of the Δ*sepH*, Δ*mlcA*, Δ*ainA*, and *sepL-1* mutants was also carried out by CRISPR/Cas9-mediated gene targeting. The complete ORF for each gene was amplified from CEA10 genomic DNA to contain regions homologous to the target sequence (5' end) as well as to a phleomycin resistance cassette (3' end). The phleomycin resistance cassette was amplified from the plasmid pAGRP [80] and primers deigned to incorporate sequence complimentary to the 3' end of the *sepL* locus. As such, an overlapping region between the two fragments was generated. New PAM sites outside of the targeted loci were selected for Cas9 RNP-targeting and both repair template fragments were mixed during transformation. Positive transformant colonies were confirmed for proper integration as described above.

## Cell wall stress and echinocandin susceptibility assays

Protein kinase disruption mutants were screened for cell wall sensitivity by monitoring their growth in the presence of cell wall disrupting agents. Qualitative primary screens were performed with at least two biological replicates of each mutant and the parental strain by arraying strains (5 μl of a $10^4$ conidia / ml suspension) onto GMM agar plates containing either 40 μg / ml or 80 μg / ml of either congo red (CR) or calcofluor white (CFW). To select positive hits from the primary screen, colony development was examined after incubation at 37°C for 72 hours and strains displaying decreased or increased colony size versus the parental isolate were chose for secondary analysis. GMM agar plates without CR or CFW were used as a growth control. For secondary analyses of strains selected from primary screens, quantitative spot dilution assays were conducted. In brief, fresh conidial suspensions were prepared from single spore isolates of each mutant and 5 μl of 10-fold serial dilutions ranging from $10^6$ to $10^3$ conidia / ml were inoculated onto GMM agar plates supplemented with 40 μg / ml or 80 μg / ml of either CR or CFW. Plates were again incubated at 37°C for 72 hours and GMM agar plates containing no compounds were used for growth control.

The *in vitro* activity of caspofungin was determined using a broth microdilution assay [50]. Briefly, ten two-fold dilutions of caspofungin, ranging from 4 to 0.0075 μg / ml, were prepared in RPMI and placed in round bottomed 96-well plates. Then, each well was inoculated with $2x10^4$ conidia and incubated at 35°C. Caspofungin minimal effective concentration (MEC) was read after 24 hours, with the aid of an inverted mirror. The strain *Candida krusei* ATCC 6258 was used as a quality control to ensure accurate activity of the tested drug. The assays were repeated at least twice for each mutant. For those mutants showing reduced or increased susceptibility by at least one two-fold dilution in comparison to the parental strain, the antifungal activity was also evaluated by a spot dilution assay. Briefly, GMM agar plates containing 0.06–0.5 μg / ml of caspofungin, were inoculated with serial dilutions of conidia suspensions ranging from $10^4$ to 10 conidia. The plates were incubated at 37°C for 72 hours and the growth of each mutant was recorded every 24 hours and compared to CEA10.

Antifungal susceptibility was also assessed using concentration gradient strips (Etest), using modification of a previously described protocol [81]. A suspension containing $10^6$ conidia in 0.5 ml was homogeneously inoculated onto GMM agar plates. Caspofungin (CAS, Biomerieux) or micafungin (MFG, Biomerieux) embedded strips were applied onto the agar and plates were incubated at 37°C. The production of a zone-of-clearance was recorded after 48 hours of culture.

## CFW and propidium iodide (PI) staining

CFW and PI staining were performed as previously described [82]. Briefly, one thousand conidia were cultured in coverslips submerged in liquid GMM. After 16 hours at 37°C, the coverslips with adherent hyphae were washed with 50 mM morpholinepropanesulfonic acid (MOPS) buffer solution, adjusted to pH 6.7 and then submerged in fixative solution (8% form-aldehyde, 25mM EGTA, 5mM $MgSO_4$, 5% DMSO and 0.2% Triton) for one hour at room temperature (RT). Coverslips were again washed twice with 50 mM PIPES for 10 min and treated with 100 μg / ml of RNAase A for one hour at 37°C. All samples were then washed twice with MOPS buffer and stained with 12.5 μg / ml of PI and 1 μg / ml of CFW for 5 min at RT. Finally, the coverslips were washed twice more with MOPS buffer, mounted and analyzed immediately by fluorescence microscopy using a Nikon Ni-U upright microscope equipped with TRITC and DAPI filters. Images were acquired using Nikon Elements software package.

## Quantitation of viability by CFDA

Viability in the presence of caspofungin was analyzed using 5,(6)-Carboxyfluorescein Diace-tate (CFDA) staining, as previously described [23]. Conidia ($8x10^4$) were inoculated into 4 ml of GMM broth with or without 0.5 μg / ml of caspofungin and poured onto sterile coverslips in 35mm petri dishes. Cultures were then incubated at 37°C to allow conidia to germinate and adhere to coverslips. At the indicated times, culture supernatants were discarded and cover-slips were stained with a solution of 50 μg / ml CFDA (Invitrogen) in 0.1M MOPS buffer (pH 3) for 1 hour at 37°C and 250rpm. Coverslips were washed once in 0.1M MOPS buffer and mounted for microscopy. Fluorescence microscopy was performed on a Nikon NiU micro-scope equipped with a Nikon DS-Qi1Mc camera using GFP filter settings. The percentage of CFDA stained microcolonies was determined by manual counts and images were captured using Nikon Elements software (v4.0).

## Animal studies

For survival studies, two different models of invasive pulmonary aspergillosis were employed, as previously described [56]. Each model utilized CF-1 female mice weighing approximately 25 g. For the corticosteroid model, mice were immunosuppressed with 40 mg / kg of triamcin-olone acetonide (TA) (Kenalog, Bristol-Myers Squibb, Princeton, NJ, USA), given subcutane-ously the day prior to the infection. For the chemotherapeutic model, mice were immune suppressed by the intraperitoneal administration of 150 mg / kg of cyclophosphamide, on days -3, +1, +4 and +7, in addition to the TA injection on day -1. On the day of the infection, mice were transiently anesthetized by the inhalation of isoflurane in an induction box (primary and secondary flow rate set at 0.5 liters / minute, 2.5% isoflurane) and inoculated by intranasal instillation with a suspension of $10^5$ (initial survival studies) or $10^6$ conidia (echinocandin ther-apy experiments) in 20 μl of saline solution. Survival and health status of the mice were moni-tored at least twice a day during a period of 15 days. Those mice showing severe signs of distress or disease were humanely euthanized by anoxia with $CO_2$ followed by cervical disloca-tion. In order to prevent bacterial infections, mice were given a mixture of sulfamethoxazole and trimethoprim in the drinking water, starting 3 days before the inoculation.

To determine the *in vivo* susceptibility to echinocandins in selected mutants, mice were immunosuppressed and inoculated as described above and, in addition, were treated with 1 or 2 mg / kg of micafungin (Mycamine, Astellas Pharma Inc., Northbrook, IL, USA) intraperito-neally, once a day. Treatments started one day post-infection (day +1) and lasted 3 days (day +4). Mock groups were given saline alone. All echinocandin treatments were based on previ-ously published studies [83]. Viable residual tissue burden in lungs of mice untreated or

treated with micafungin was examined 6 hours (day 0) and 4 days (day +4) after infection. Mice were euthanized at the indicated time, lungs were aseptically harvested, sectioned into small pieces, and cultured on yeast peptone dextrose (YPD) agar for 48 hours at 37˚C.

In addition to the survival studies, histopathology analyses were performed. Two mice per group were immunosuppressed and infected as described above and euthanized 4 days after the inoculation. Lungs were inflated by intratracheal perfusion with 10% buffered formalin and subsequently embedded in paraffin. Finally, multiple 5 μm sections from the superior, middle and inferior lobes were stained with Grocott's methenamine silver for visualization of fungal elements.

## Lung fungal burden and *in vivo* cytokine secretion

Lung fungal burden by quantitative PCR (qPCR) and measurement of *in vivo* cytokine secretion by ELISA were performed using slight modifications of previously described protocols [57,84]. Mice immune suppressed following the chemotherapeutic model described above were intranasally inoculated with $10^6$ conidia. After 4 days of infection, mice were euthanized and lungs were harvested and homogenized in 1 ml of sterile PBS using the gentleMACS dissociator (Miltenyi Biotec). Approximately 700 μl of lung homogenate were lyophilized and processed for DNA extraction using the E.Z.N.A. Fungal DNA Mini Kit (Omega Bio-tek) according to manufacturer instructions. qPCR analyses were performed in technical duplicate for each sample, using the PrimeTime Gene Expression Master Mix and qPCR Probe Assays (Integrated DNA Technologies) containing the primers to amplify a region of the *A. fumigatus* 18S rRNA gene, as previously described [85]. For each sample, 500 ng of total DNA was used as template, and a standard curve containing 100, 10, 1, 0.1 and 0.01 ng of CEA10 genomic DNA was also included in the assay so that the amounts of *A. fumigatus* specific DNA could be determined. qPCR was conducted on a Bio-Rad CFX96 Real-Time PCR system running the Bio-Rad CFX Maestro 1.0 software (v4.0). Data are represented as nanograms of *A. fumigatus* specific DNA in 500 ng of total DNA.

For analysis of cytokine secretion in the lungs, 300 μl of lung homogenate were mixed with 2x Protease Inhibitor Cocktail containing AEBSF, Aprotinin, Bestatin, E-64, Leupeptin and Pepstatin A (Sigma) and centrifuged at 14000 rpm for 10 minutes. Resulting supernatants were kept at -80˚C until employed for further measurements using mouse TNFα and IL-1β ELISA kits (Invitrogen), according to manufacturer instructions. Total protein quantification was performed using Quick Start Bradford Protein Assay (Bio-Rad) and used for cytokine level normalization.

## Analysis of cytokine release in THP-1 cells

WT (THP1-null), *Nlrp3*$^{-/-}$ (THP1-KO-NLRP3), and *Asc*$^{-/-}$ (THP1-KO-ASC) THP-1 cells (Invivogen) were utilized. Cytokine release from THP-1 cells was performed as previously described, with minor modifications [59,86]. Briefly, cells were cultured in RPMI-1640 medium containing 25 mM HEPES supplemented with 10% heat-inactivated Fetal Bovine Serum, 100 U/mL penicillin-streptomycin, and 100 μg/mL normocin as described previously. THP-1 cells were assessed for viability by exclusionary Trypan Blue staining and plated at a density of $10^5$ cells/well in 96-well microtier plates using similar medium lacking normocin. Phorbol 12-myristate 13-acetate (PMA) was added at 100 nM final concentration and cells incubated for 24h to adopt a macrophage phenotype. Supernatants were discarded and replaced with 180 μl of fresh RPMI (without phenol red) and 20 μl of distilled water containing $10^6$ *A. fumigatus* conidia (MOI 10:1) added. In some cases, the inflammasome inhibitor MCC950 (10 μM, Invivogen) was added to the THP-1 cells concomitantly with the conidia.

THP-1 cells and conidia were co-cultured for 16h and supernatants were collected and analyzed by ELISA for IL-1β concentration following manufacturer instructions.

## Supporting information

**S1 Fig. Disruption of multiple protein kinase genes results in reduced colony growth and conidiation of A. fumigatus. A)** Quantitation of colony diameters for the wild type parent strain (CEA10) and disruption mutants displaying minimal, moderate or severe growth restriction on minimal media. 10,000 conidia from each strain were point-inoculated onto minimal media and cultured for 96 hrs at 37˚C. Colony diameters form triplicate cultures for each strain were measured (mm) and averaged. Statistical comparisons were made by ANOVA and all comparisons generated a p $\leq$ 0.0371. Disruption mutants not shown generated colony diameters that were similar to CEA10. **B)** Quantitation of conidiation for the parent strain (CEA10) and multiple protein kinase gene disruption mutants. Conidia ($2 \times 10^4$) from each strain were cultured as in (A). Colony area was calculated and conidia were harvested in 10 ml of sterile water before filtration and quantitation using a hemocytometer. Each strain was assayed in triplicate and data were averaged. Statistical comparisons were made by one-way ANOVA with Dunnett's multiple comparisons post hoc and all comparisons generated a p $\leq$ 0.0036. Disruption mutants not included here for colony diameter or conidiation analyses were not significantly different from the parental strain (CEA10).
(TIF)

**S2 Fig. Gene deletion of sepH and gene reconstitution of the sepL disruption (sepL-1).** Schematics for deletion of *sepH* (**A**) and for complementation of the *sepL-1* disruption mutant (**C**). Genetic manipulations were carried out using CRISPR/Cas9 gene editing (see *Materials and Methods*). For each locus targeted, the 20-nucleotide protospacer (black font) and the 3-nucleotide protospacer adjacent motif (PAM, underlined red font) are displayed. Each manipulation utilized Cas9-mediated double strand breaks generated 5' and 3' of the targeted gene. Repair templates (HygR = hygromycin resistance cassette; PhleoR = phleomycin resistance cassette) were PCR amplified from plasmids using primers that incorporated 40-basepair microhomology arms for targeting. Correct integration of repair templates was confirmed by PCR using primers P1 and P2 for *sepH* deletion (**B**) and primers P3 and P4 for *sepL-1* complementation (**D**).
(TIF)

**S3 Fig. Deletion and complementation of A. fumigatus genes AFUB_067160, encoding myosin light chain (mlcA), and AFUB_055850, encoding alpha-actinin (ainA).** Schematics for deletion (**A** and **D**) and complementation (**B** and **E**) of *mlcA* and *ainA*, respectively. Genetic manipulations were carried out using CRISPR/Cas9 gene editing (see *Materials and Methods*). For each locus targeted, the 20-nucleotide protospacer (black font) and the 3-nucleotide protospacer adjacent motif (PAM, underlined red font) are displayed. Each manipulation utilized Cas9-mediated double strand breaks generated 5' and 3' of the targeted gene. Repair templates (HygR = hygromycin resistance cassette) were PCR-amplified from plasmids using primers that incorporated 40-basepair microhomology arms for targeting. Correct integration of repair templates and gene complementations were confirmed by PCR using primers P5 and P6 for *mlcA* (**A** and **B**) and primers P7 and P8 for *ainA* (**E** and **F**).
(TIF)

**S4 Fig. Gene disruption of A. fumigatus cell wall integrity kinases (mkk2 and mpkA) and the protein kinase A catalytic subunit (pkaC1) generates increased susceptibility to echinocandins.** Modified E-test assays for the *mkk2-1*, *mpkA-1*, and *pkaC1-1* disruption mutants

using minimal media (see *Materials and Methods*). Note the residual growth in the zone-of-clearance for both *mkk2-1* and *mpkA-1* mutants indicating lack of echinocandin cidal activity. Insets show representative, drug-free minimal media culture plates onto which a single agar plug from the zone-of-clearance for each assay was sub-cultured. Multiple agar plugs (n = 10), taken from within 1 cm of the E-test strip and between the 32 and 0.25 μg/ml markers, were sub-cultured in the same manner for each assay. CAS = caspofungin; MFG = micafungin. (TIF)

**S1 File.**
(XLSX)

## Author Contributions

**Conceptualization:** Brian M. Peters, Jarrod R. Fortwendel.

**Formal analysis:** Ana Camila Oliveira Souza, Adela Martin-Vicente, Jarrod R. Fortwendel.

**Funding acquisition:** Jarrod R. Fortwendel.

**Investigation:** Ana Camila Oliveira Souza, Adela Martin-Vicente, Ashley V. Nywening, Wenbo Ge, David J. Lowes, Brian M. Peters, Jarrod R. Fortwendel.

**Methodology:** Ana Camila Oliveira Souza, Adela Martin-Vicente, Ashley V. Nywening, Wenbo Ge, David J. Lowes, Brian M. Peters, Jarrod R. Fortwendel.

**Project administration:** Jarrod R. Fortwendel.

**Supervision:** Jarrod R. Fortwendel.

**Writing – original draft:** Ana Camila Oliveira Souza, Adela Martin-Vicente, Brian M. Peters, Jarrod R. Fortwendel.

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
