## [Decision Letter · Decision Letter 0]

28 Jun 2021

Dear Dr Fortwendel,

Thank you very much for submitting your manuscript "Loss of Septation Initiation Network (SIN) kinases blocks tissue invasion and unlocks echinocandin cidal activity against Aspergillus fumigatus" for consideration at PLOS Pathogens. As with all papers reviewed by the journal, your manuscript was reviewed by members of the editorial board and by several independent reviewers. The reviewers appreciated the attention to an important topic. Based on the reviews, we are likely to accept this manuscript for publication, providing that you modify the manuscript according to the review recommendations.

Specifically, all three reviewers appreciated the novelty and impact of the work on our understanding of kinases involved in A. fumigatus pathogenesis and interactions with the echinocandins. While several interesting and informative experiments around the specific impact of the SIN kinase mutants on the phenotypes observed are suggested by all 3 reviewers, I will leave it to you to decide whether you'd like to address these experimentally or through a softening of the conclusions drawn based on the current data. The manuscript contains significant new data and a new resource for the community. It is clear that the excellent reviewer suggestions will strengthen our mechanistic understanding of the phenotypes observed, but (at your discretion) they may be beyond the scope of the current study given the nature of the suggested experiments, particularly those related to the cell wall and kinase molecular function.

Sincerely,

Robert A. Cramer, PhD

Associate Editor

PLOS Pathogens

Alex Andrianopoulos

Section Editor

PLOS Pathogens

Kasturi Haldar

Editor-in-Chief

PLOS Pathogens

orcid.org/0000-0001-5065-158X

Michael Malim

Editor-in-Chief

PLOS Pathogens

orcid.org/0000-0002-7699-2064

Reviewer Comments (if any, and for reference):

Reviewer's Responses to Questions

**Part I - Summary**

Reviewer #1: This ms uses CRISPR/Cas9-mediated gene targeting to generate a library of protein kinase disruption mutants in A. fumigatus. The library was then challenged with cell wall stress agents (CR, CFW and caspofungin) in order to identify those kinases that participate in adaptive responses to loss of cell wall integrity. Several mutants were identified that confirmed what is already known in the literature, nicely demonstrating the validity of this mutation/screening strategy. The focus of the subsequent analysis was on three kinase genes, comprising a tripartite kinase cascade known as the septation initiation network (SIN). The data reveal that deletion of any of these three genes abrogates hyphal septation and increases susceptibility to cell wall stress. Importantly, this correlated with extensive hyphal damage in response to treatment with echinocandins, as well as a striking loss of invasive growth potential in two immunologically distinct mouse models of infection. By analyzing other non-kinase regulators of septation the study provides evidence that these phenotypes are a likely consequence of septation deficiency rather than SIN-kinase dependent activity. The study also demonstrates that echinocandin therapy is effective at eliminating residual fungal burden in mice infected with septation-deficient mutants, providing proof of principle that septation inhibitors would be effective in combination therapy with the echinocandin class.

The manuscript was very well written, the data are compelling, and the conclusions are supported by the findings. The major strength of the paper is that the study provides new evidence that A. fumigatus relies on septation for invasive growth and that the efficacy of echinocandin therapy could be amplified by septation inhibitors.

Reviewer #2: This impressive and interesting study describes the authors’ efforts to characterize protein kinases of A. fumigatus on a comprehensive scale and to scrutinize the role of a selected set of kinases in the context of antifungal resistance adaptation. More specifically, 118 disruption mutants in genes encoding conserved protein kinases were generated and comprehensively screened for growth, fitness, and, mostly relevant, minimum effective concentrations of the fungal cell wall synthesis inhibitors of the echinocandin class. Emerging from these activities, two kinases presumed to serve as core components of the SIN cascade were characterized in depth to reveal their role in septation and virulence. In essence, the importance of septation within the fungal fungal hypha to counter cell wall stress and to support invasive growth within a susceptible host could be demonstrated.

Reviewer #3: This is a well written and interesting study that presents an impressive amount of work demonstrating the importance of Aspergillus fumigatus SIN kinases in virulence as well as "resistance" to the cidal effects of echinocandins. The experiments are well designed, the data is convincing and the paper is well written an thoughtful.

My only real comment is understanding the link between virulence/resistand and septation per se vs septation-associated biological processes. While I would like to believe that septation is what saves Aspergillus fumigatus from candins, and the data presented is certainly is compatible with this, we know that C. albicans can form septae, yet is highly susceptible to the cidal activity of these agents - so it seems that septae are not a guaranteed way to resist these agents. The obvious question that remains unanswered is - what are the effects of these deletions on the composition of the cell wall other than affecting septation? It looks to me like the sepH mutant has increased CFW staining, and the altered sensitivity to other cell wall agents suggests there are fundamental changes to the cell wall for these mutants.

**Part II – Major Issues: Key Experiments Required for Acceptance**

Reviewer #1: I realize that the focus is on echinocandins, but I think the echinocandin results really beg the question about other antifungal classes. Is there any hypersensitivity to nikkomycin or triazoles?

Reviewer #2: Given the plethora of novel and relevant data, relevance of this manuscript is clearly seen by this reviewer. Two interlinked scientific parts are presented, the rather shallow characterization of more than 100 protein kinase mutants that had been generated by an innovative semi-high-throughput approach, to be followed by extensive phenotyping of putative SIN kinase disruptants and regulators of hyphal septation. To fully exploit the range of insights that might be gained from the presented mutants and their phenotypes, some further lines of research could be followed and experimental tasks executed, such as biochemical studies on the presumed kinase targets and their phosphorylation or exploring the postulated positive effect of suited kinase inhibitors or septum formation inhibitors on echinocandin activity (static vs. cidal) and therapy. Yet, the most interesting question arises from the striking phenotype of avirulence that is displayed by the SIN kinase mutants upon pulmonary infection of susceptible mice: Presuming that this results from hampered tissue invasion, one wonders about the outcome in a systemic infection setting…

Reviewer #3: I would suggest doing an analysis of the cell wall glycan composition of these mutants (with and without echinocandins), at least to confirm if the standard cell wall integrity responses are compromised, and if there are fundamental changes in glycans that could contribute to the observations in the paper. If this is not done, I would suggest a careful rewrite of the discussion to emphasize that while clearly the SIN kinases are required for all the observed phenotypes (and dramatically so!), the current experiments do not distinguish between septation per se, and other SIN kinase-dependent process, as potential mechanisms underlying the findings.

**Part III – Minor Issues: Editorial and Data Presentation Modifications**

Reviewer #1: It is interesting that similar mutants in a plant pathogen reduce turgor pressure necessary for appressoria, raising the possibility that the lack of invasiveness is a related problem. Will the mutants grow on explants of lung tissue tissue on plates similar to how they behave on minimal medium? If they can’t invade the substrate, perhaps not. This could get complicated, so perhaps experiments along these lines would be best left to a follow-up study that delves into the invasive properties of the hyphae.

Are the mutants temperature sensitive? This is not a major issue since the focus of the ms is on pathogenesis and hence 37oC is the most relevant. However, I would be curious to see it mentioned in the discussion if the results are already available.

Line 67 – is limited by patient toxicity and threatened by resistance

Reviewer #2: Besides these rather general points, a couple of other, more specific ones exist that might be addressed before considering this fine manuscript for publication in PLoS Pathogens; in detail, these are as follows:

Significant parts of the Introduction are spent on describing results (e.g., l. 99-106 or l. 121-131). Maybe providing more context with respect to invasive(!) aspergillosis and knowledge about virulence determinants thereof might be illustrative without giving away too many findings. In the Results section, it is puzzling why the disruption library was established in the CEA10 wild-type isolate after identifying kinase-encoding orthologues in the genome sequences of A1163 and Af293 (l. 137-145) – please clarify for the readers that are not familiar with A. fumigatus laboratory isolate lineages. What might be the reason for the unexpected observation that several protein kinase mutants display a mild growth reduction that has not, with the exception of bck1, been described for any of the corresponding A. fumigatus mutants before (l. 185-193)? What is the actual mode of action for the cell wall stressors Calcofluor White and Congo Red (l. 208/9)? The notion that some of the library strains might not be altered in the respective gene function is puzzling (l. 219/202) – to what extent was this checked for the bck1 gene that gave rise to this suspicion albeit displaying a growth phenotype? Hyphal damage of SIN kinase mutants in the presence of micafungin is demonstrated by PI permeability assays (l. 319-330), which could be further supported by leakage experiments using strains that express fluorescent proteins. Description of the virulence phenotypes of SIN kinase mutants (l. 349-352) is somehow unclear. When confirming dependency of IL-1beta secretion on NLRP3 inflammasome activity (l. 380-385), citing one of the most recent literature on this rather distractive issue (e. g., PMID: 33268895) might be convenient. The Discussion’s reasoning that viable fungal elements are especially problematic for neutropenic patients (l. 556-557) is somehow awkward. When speculating on the lack of turgor pressure in the tip compartment being the underlying mechanisms for the observed virulence and tissue invasion phenotypes, recent insights deduced from microfluidic experimentations (PMID: 33727355) could be incorporated.

Reviewer #3: none

PLOS authors have the option to publish the peer review history of their article (what does this mean?). If published, this will include your full peer review and any attached files.

Reviewer #1: No

Reviewer #2: No

Reviewer #3: No

Figure Files:

Data Requirements:

Reproducibility:

References:

---

## [Editor Report · Decision Letter 1]

16 Jul 2021

Dear Dr Fortwendel,

We are pleased to inform you that your manuscript 'Loss of Septation Initiation Network (SIN) kinases blocks tissue invasion and unlocks echinocandin cidal activity against Aspergillus fumigatus' has been provisionally accepted for publication in PLOS Pathogens.

Best regards,

Robert A. Cramer, PhD

Associate Editor

PLOS Pathogens

Alex Andrianopoulos

Section Editor

PLOS Pathogens

Kasturi Haldar

Editor-in-Chief

PLOS Pathogens

orcid.org/0000-0001-5065-158X

Michael Malim

Editor-in-Chief

PLOS Pathogens

orcid.org/0000-0002-7699-2064
---

## [Editor Report · Acceptance letter]

4 Aug 2021

Dear Dr Fortwendel,

We are delighted to inform you that your manuscript, "Loss of Septation Initiation Network (SIN) kinases blocks tissue invasion and unlocks echinocandin cidal activity against Aspergillus fumigatus," has been formally accepted for publication in PLOS Pathogens.

Best regards,

Kasturi Haldar

Editor-in-Chief

PLOS Pathogens

orcid.org/0000-0001-5065-158X

Michael Malim

Editor-in-Chief

PLOS Pathogens

orcid.org/0000-0002-7699-2064